# ALLO-1- and IKKE-1-dependent positive feedback mechanism promotes the initiation of paternal mitochondrial autophagy

Taeko Sasaki [1,2], Yasuharu Kushida[2], Takuya Norizuki [1], Hidetaka Kosako [3], Ken Sato [2] ✉ & Miyuki Sato [1] ✉

Allophagy is responsible for the selective removal of paternally inherited organelles, including mitochondria, in *Caenorhabditis elegans* embryos, thereby facilitating the maternal inheritance of mitochondrial DNA. We previously identified two key factors in allophagy: an autophagy adaptor allophagy-1 (ALLO-1) and TBK1/IKKε family kinase IKKE-1. However, the precise mechanisms by which ALLO-1 and IKKE-1 regulate local autophagosome formation remain unclear. In this study, we identify two ALLO-1 isoforms with different substrate preferences during allophagy. Live imaging reveals a stepwise mechanism of ALLO-1 localization with rapid cargo recognition, followed by ALLO-1 accumulation around the cargo. In the *ikke-1* mutant, the accumulation of ALLO-1, and not the recognition of cargo, is impaired, resulting in the failure of isolation membrane formation. Our results also suggest a feedback mechanism for ALLO-1 accumulation via EPG-7/ATG-11, a worm homolog of FIP200, which is a candidate for IKKE-1-dependent phosphorylation. This feedback mechanism may underlie the ALLO-1-dependent initiation and progression of autophagosome formation around paternal organelles.

Mitochondria are semi-autonomous organelles with their own DNA, known as mitochondrial DNA (mtDNA). Unlike nuclear DNA, mtDNA was discovered in the 1960s to be inherited from one parent in the fungus *Neurospora crassa*[1–3]. Such uniparental inheritance occurs in most eukaryotes, including mammals[3–5]. Although this inheritance is a universal phenomenon, the mechanism and biological significance of uniparental inheritance of mtDNA remain unclear. Recent studies showed that paternal mitochondria and mtDNA are actively eliminated before and after fertilization in various species[5]. In the nematode *Caenorhabditis elegans*, mtDNA is inherited maternally, similar to in mammals, including humans. After fertilization, the paternal mitochondria with their mtDNA enter the oocyte and are specifically eliminated via selective autophagy (Fig. 1a)[6–8]. Autophagy also degrades other sperm-derived organelles, named as membranous organelles (MOs; sperm-specific post-Golgi organelles), which are essential for sperm fertility[9]. This autophagic degradation of sperm-derived organelles is termed as allophagy (allogeneic [non-self] organelle autophagy)[10,11]. Degradation of paternal mitochondria via autophagy occurs in *Cryptococcus neoformans*, *Drosophila melanogaster*, and mice[12–14], suggesting that this process is widely conserved across different species.

In macroautophagy (hereafter referred to as autophagy), a portion of the cytoplasm is sequestered nonselectively or selectively by double-membrane autophagosomes and targeted to lysosomes for degradation. This process is mediated by autophagy-related (Atg) proteins that are highly conserved from yeast to mammalian cells[15,16].

[1]Laboratory of Molecular Membrane Biology, Institute for Molecular and Cellular Regulation, Gunma University, Maebashi, Gunma 371-8512, Japan. [2]Laboratory of Molecular Traffic, Institute for Molecular and Cellular Regulation, Gunma University, Maebashi, Gunma 371-8512, Japan. [3]Division of Cell Signaling, Fujii Memorial Institute of Medical Sciences, Institute of Advanced Medical Sciences, Tokushima University, Tokushima 770-8503, Japan. ✉e-mail: sato-ken@gunma-u.ac.jp; m-sato@gunma-u.ac.jp

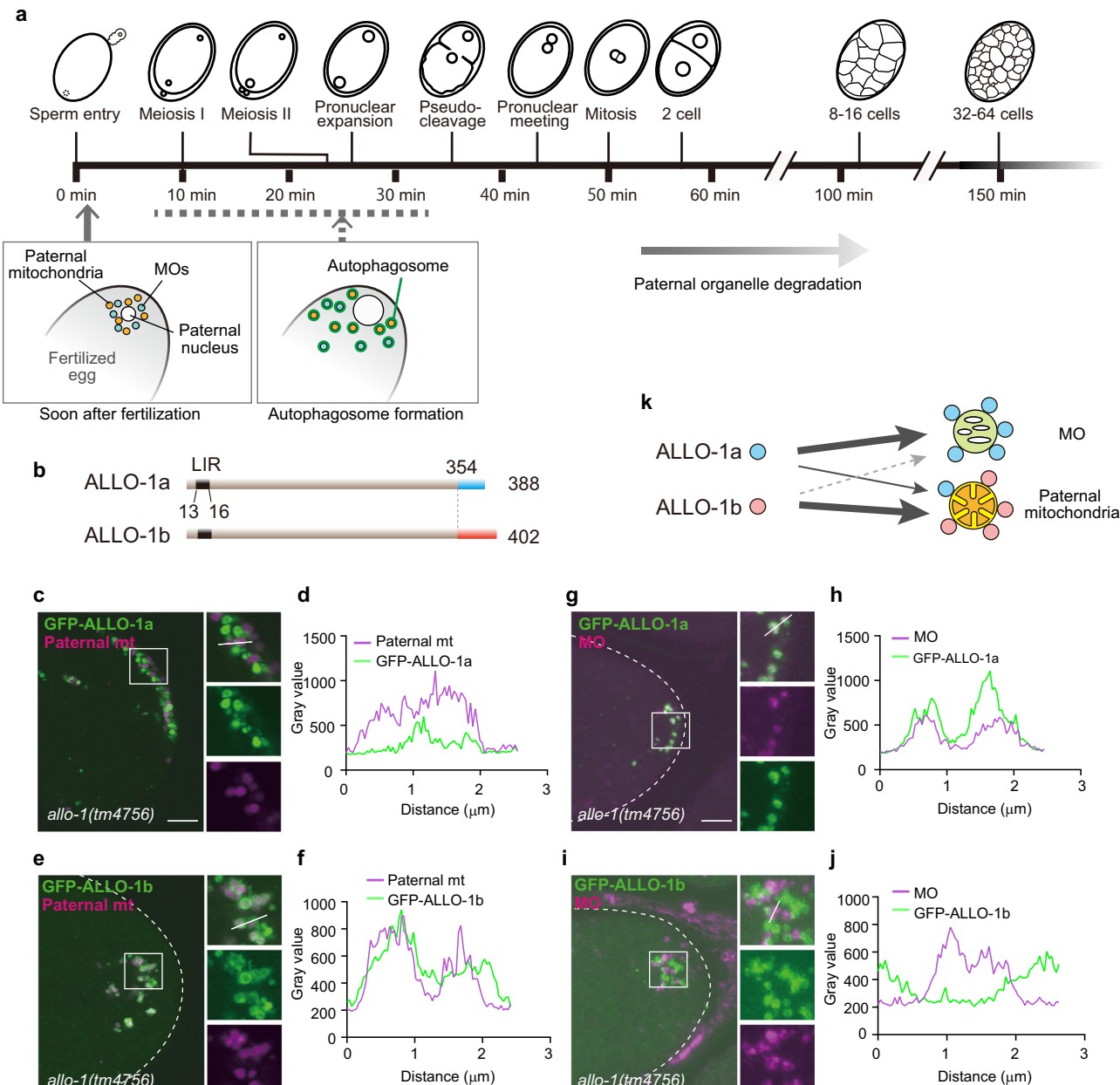

**Fig. 1 | Localization of allophagy-1 (ALLO-1) isoforms during allophagy. a** Time course of allophagy and embryogenesis previously reported[37]. The general time course of early embryogenesis at 20 °C is shown in the frame. The exact time of autophagosome formation was not clear (dotted line). **b** Structure of ALLO-1a (top) compared with that of ALLO-1b (bottom). The color of areas with different sequences is indicated by cyan or orange bars. The positions of LC3-interacting region motifs (LIRs) are shown (dark gray bars). **c**–**f** Localization of ALLO-1 isoforms around paternal mitochondria in the *allo-1* mutant background. Fertilized eggs at 1-cell stage (hereafter referred to as zygotes) dissected from adult hermaphrodites expressing green fluorescent protein (GFP)-ALLO-1a or b (green) under the oocyte-specific *pie-1* promoter and HSP-6-mCherry under the sperm-specific *spe-11* promoter (paternal mitochondria (mt); magenta). Plot profiles of white lines in the upper right panels of **c** and **e** are shown in **d** and **f**, respectively. **g**–**j** Localization of ALLO-1 isoforms around the membranous organelles (MOs) in the *allo-1* mutant background. Zygotes dissected from adult hermaphrodites expressing GFP-ALLO-1a or b (green) were stained with a monoclonal antibody, 1CB4, which recognizes MOs (magenta). Plot profiles of white lines in the upper right panels of **g** and **i** are shown in **h** and **j**, respectively. In total, 67 (**c**, **d**), 69 (**e**, **f**), 37 (**g**, **h**), and 32 (**i**, **j**) zygotes were observed, with similar patterns observed in all zygotes. All zygotes were observed between meiosis II and pseudo-cleavage stage. Outline of the zygote is indicated by dotted white line. Scale bars, 5 μm. **k** Schematic representation of localization pattern of ALLO-1 isoforms.

Atg1/UNC-51-like autophagy-activating kinase (ULK) complex controls the initiation of autophagy, and ordered recruitment of downstream factors leads to de novo formation of an isolation membrane that finally forms autophagosomes. In selective autophagy pathways, cargos such as protein aggregates and damaged organelles are selectively engulfed by autophagosomes via molecules known as autophagy receptors/adaptors[17,18]. Autophagy receptors/adaptors reside on or localize to the cargos recognizing the "eat me" signal on the cargos, which typically involves the ubiquitination of the cargos. These receptors/adaptors contain a motif named as the LC3-interacting region (LIR), which directly binds to autophagic regulator Atg8/LC3 family proteins[19]. As Atg8/LC3 proteins are anchored to the autophagosomal membrane, this interaction has long been considered as a mechanism for selective engulfment of cargos[20]. However, recent studies revealed that a subset of mammalian autophagy adaptors directly binds to FIP200, a scaffold protein of the ULK complex, and

regulates the recruitment of this complex[21–25], suggesting that autophagy adaptors directly control autophagy initiation on the cargo. In mammals, TRAF-associated NF-κB activator-binding kinase 1 (TBK1) is required for several selective autophagy pathways, including the PTEN-induced kinase 1 (PINK1)–Parkin-mediated mitophagy. TBK1 phosphorylates several autophagy adaptors such as OPTN, NDP52, and p62, and promotes binding to LC3 or ubiquitin[26–29]. Additionally, TBK1-mediated phosphorylation of LC3 family proteins or Rab7A is involved in PINK1–Parkin dependent mitophagy[30,31]. A recent study also suggested that TBK1 directly binds to class III phosphatidylinositol 3-kinase to initiate autophagy in the OPTN-mediated mitophagy, whereas it functions redundantly with ULK1/2 in the NDP52-mediated pathway[32]. Thus, TBK1 has been suggested to play multiple roles; however, the mechanism by which TBK1 regulates selective autophagy is not fully understood, particularly in animal models and systems other than mammals.

We previously reported two essential factors for allophagy, allophagy-1 (ALLO-1) and inhibitor of NF-κB kinase epsilon subunit homolog-1 (IKKE-1)[33]. ALLO-1 is a cytosolic protein containing the LIR motif and no known functional domains such as a mitochondrial targeting signal or transmembrane domain. ALLO-1 resides in the cytoplasm of oocytes and selectively localizes to paternal organelles after fertilization. Although ALLO-1 is only conserved in nematodes, it directly binds to LC3, GABARAP, and GATE-16 family-1 (LGG-1), the worm homolog of Atg8/LC3, via the LIR motif, and functions as an autophagy adaptor. IKKE-1, the worm homolog of mammalian TBK1 and IKKε family kinases, was identified as a binding partner of ALLO-1. In allo-1 or ikke-1 deletion mutants, the degradation of paternal organelles fails, and paternal mtDNA is inherited by the next generation[33]. However, the mechanism by which these factors control local autophagosome formation around the cargo and the specific role of IKKE-1 phosphorylation remain unclear.

In this study, we observed that ALLO-1 has two isoforms with different C-terminal sequences, ALLO-1a and ALLO-1b, which exhibit distinct cargo preference during allophagy. Live imaging analysis revealed a stepwise mechanism of ALLO-1 localization involving rapid cargo recognition and subsequent ALLO-1 accumulation on the cargo. This accumulation step was regulated by IKKE-1-dependent phosphorylation. Moreover, ALLO-1 directly binds to ectopic P granules-7 (EPG-7)/autophagy-related-11 (ATG-11), the worm FIP200 homolog, and mediates recruitment of the ULK complex around cargo. Accumulation of ALLO-1 on the cargo also depends on EPG-7/ATG-11, suggesting a positive feedback mechanism promoting ALLO-1 accumulation. This feedback mechanism may promote the initiation and progression of local autophagosome formation around the paternal organelles.

## Results

### ALLO-1 has two isoforms with distinct cargo preference

ALLO-1 has an LIR motif in the N-terminal region; however, no other functional motifs were observed in our previous database search[33]. Through a database search using WormBase[34], we observed that ALLO-1 has two splice variants with different C-terminal sequences: ALLO-1a and ALLO-1b (Fig. 1b). In our previous study, a genomic fragment of allo-1, which potentially expresses both isoforms, was used in most experiments. Because it has been suggested that the C-terminal half of ALLO-1 is important for its localization to paternal organelles[33], we first investigated the localization patterns of these isoforms in fertilized eggs by expressing green fluorescent protein (GFP)-tagged ALLO-1a or ALLO-1b in the germline of the allo-1 deletion mutant. The paternal mitochondria were visualized using mCherry-fused HSP-6, a mitochondrial matrix protein, expressed under the sperm-specific spe-11 promoter[33]. MOs were immunostained using an anti-MO antibody[35,36]. Because the fluorescence intensity of GFP-ALLO-1 around paternal organelles gradually increases and is highest from meiosis II to the

pronuclear meeting stage[33], zygotes around this stage were observed. ALLO-1a was detected around both paternal mitochondria and MOs. However, a brighter signal was observed for MOs than for paternal mitochondria (Fig. 1c, d, g, h; Supplementary Fig. 1). In contrast, ALLO-1b predominantly localized around paternal mitochondria (Fig. 1e, f, i, j). These differences in the localization patterns were confirmed using immunostaining with ALLO-1a- or ALLO-1b-specific antibodies (Supplementary Fig. 2a–h). The specificity of these antibodies was evaluated using immunostaining and immunoblotting (Supplementary Fig. 3). The results showed that both isoforms are endogenously expressed in the germline. We also investigated whether the ALLO-1a or b specific region is sufficient for the localization. When the C-terminal region of ALLO-1a (355–388) or b (355–402) was fused to GFP, both fusions failed to localize to the paternal organelles and showed a diffuse pattern (Supplementary Fig. 2i, j). This suggests that the common region is also involved in localization or proper protein folding. We further examined whether these isoform-specific transgenes can rescue allophagy defects in allo-1 mutants. In the wild type, paternal organelles are sequestered by autophagosomes during the 1-cell stage and gradually disappear by the 16-cell stage (Fig. 1a)[6]. We observed that the expression of GFP-ALLO-1a rescued the defects in the allo-1 mutant during MO degradation (Fig. 2c, d). However, degradation of paternal mitochondria was delayed in embryos expressing GFP-ALLO-1a compared to that in the wild-type control, and certain paternal mitochondria were still observed in embryos at the 32–64-cell stages, suggesting that ALLO-1a is partially involved in paternal mitochondrial degradation (Fig. 2a, b). In contrast to ALLO-1a, the expression of GFP-ALLO-1b efficiently rescued the defective degradation of paternal mitochondria (Fig. 2a, b). The expression of GFP-ALLO-1b slightly restored the defect of MO degradation (Fig. 2c, d). Although GFP-ALLO-1b was not detected on MOs under our experimental conditions, it may be weakly involved in MO degradation. Transgene expressing the two isoforms rescued the degradation defects in both paternal organelles (Fig. 2b, d). The results of these rescue experiments were consistent with the localization patterns of ALLO-1 isoforms, suggesting that ALLO-1a and ALLO-1b have distinct cargo preferences during allophagy (Fig. 1k).

### ALLO-1 and IKKE-1 rapidly localize on paternal mitochondria

To examine how autophagy initiation is controlled by ALLO-1 and IKKE-1, we performed time-lapse imaging of allophagy in living worms. In C. elegans gonads, mature oocytes ovulate into the spermatheca, and the ovulated oocyte is fertilized immediately after sperm–oocyte contact (Supplementary Fig. 4)[37]. With time-lapse imaging, it was difficult to define the time of fertilization, as the cell membrane was not labeled. Therefore, the frame of sperm–oocyte contact was defined as 0 s.

We first observed the time course of autophagosome formation using the autophagosome marker GFP-LGG-1. GFP-LGG-1 appeared as small puncta close to paternal mitochondria labeled with HSP-6-mCherry at approximately 5–12 min after sperm–oocyte contact. The fluorescence intensity of GFP-LGG-1 gradually increased and the paternal organelles were surrounded by GFP-LGG-1 for 20 min (Fig. 3a; Supplementary Movie 1).

A previous study showed that ALLO-1 and IKKE-1 localize to paternal organelles before LGG-1 recruitment[33]. Time-lapse imaging revealed that GFP-ALLO-1a was detected on paternal mitochondria within 30 s, and then on structures outside the paternal mitochondria approximately 3 min after sperm–oocyte contact (Fig. 3b; Supplementary Movie 1). The fluorescence intensity of these extra-mitochondrial ALLO-1a puncta further increased and eventually became brighter than that of the paternal mitochondria. These structures were considered as MO-associated ALLO-1a based on the localization pattern of ALLO-1a (Fig. 1). In contrast, GFP-ALLO-1b was detected around the paternal mitochondria within 30 s of sperm–oocyte contact, and its signal increased predominantly on the

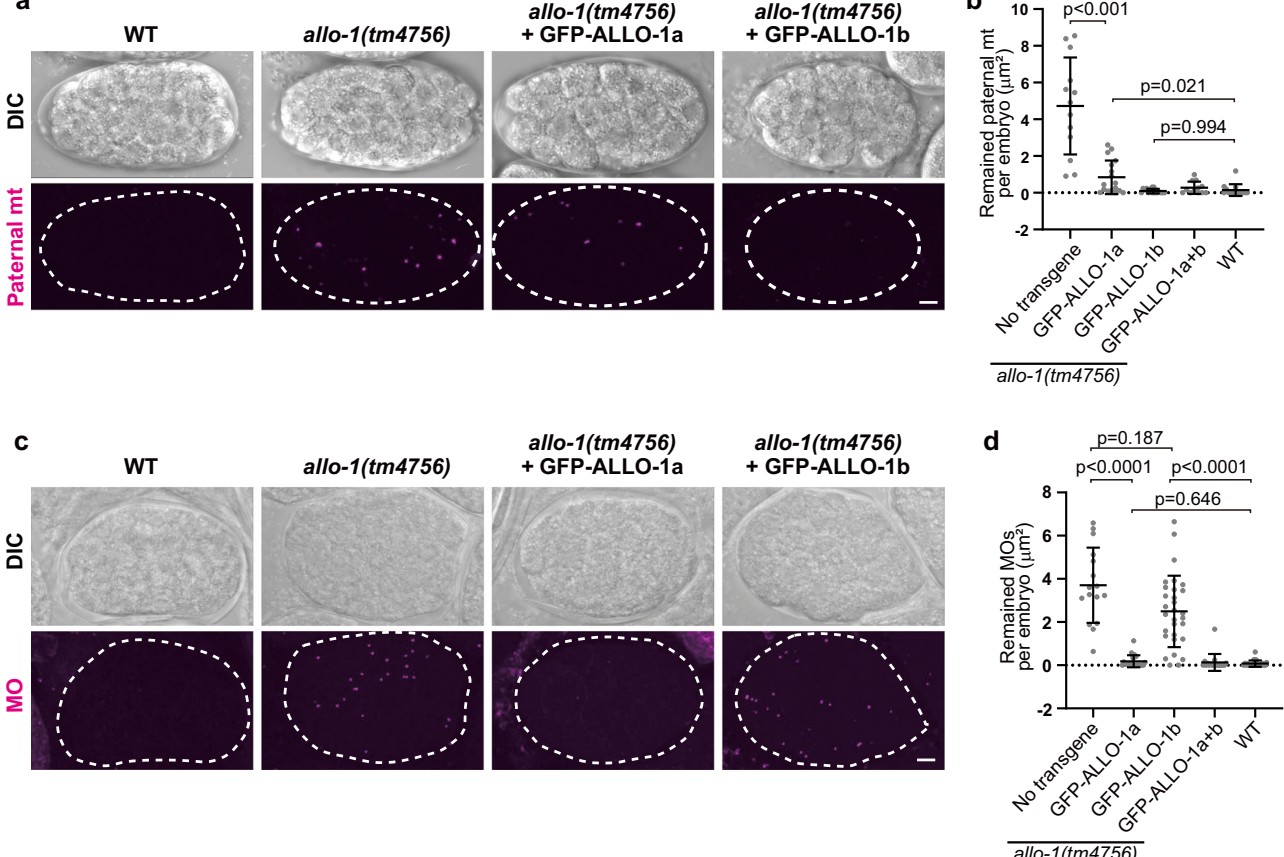

**Fig. 2 | Distinct cargo preference of ALLO-1 isoforms during allophagy.**
**a**, **c** Rescue experiment of the degradation of paternal mitochondria (mt) (**a**) or membranous organelles (MOs) (**c**). 32–64-cell stage embryos from the wild type (WT), the *allo-1* mutant without a transgene or the *allo-1* mutant expressing GFP-ALLO-1a or b were observed for paternal mitochondria (HSP-6-mCherry) and MOs (immunostaining with the antibody 1CB4). **b** Paternal mitochondria (HSP-6-mCherry) remaining in the 32–64-cell stage embryos were quantified as the area showing a fluorescent signal per embryo. *n* = 13 (no transgene), *n* = 17 (GFP-ALLO-1a), *n* = 16 (GFP-ALLO-1b), *n* = 12 (GFP-ALLO-1a+b), and *n* = 14 (WT) embryos. **d** MOs

remaining in the 32–64-cell stage embryos were quantified as the area showing a fluorescent signal per embryo. *n* = 16 (no transgene), *n* = 23 (GFP-ALLO-1a), *n* = 30 (GFP-ALLO-1b), *n* = 18 (GFP-ALLO-1a+b), and *n* = 23 (WT) embryos. Error bars represent the mean ± standard deviation (SD). Source data are provided as a Source Data file. Kruskal–Wallis with Steel-Dwass pairwise comparison test was performed for *p* values. *p* values are provided in the figures and source data file. To define the embryonic stage, differential interference contrast (DIC) images are also shown in **a** and **c**. Eggshells are indicated by dotted white lines. Scale bars, 5 μm.

paternal mitochondria (Fig. 3c; Supplementary Movie 1). Rapid localization was observed for GFP-IKKE-1 (Fig. 3d; Supplementary Movie 1). The fluorescence intensities of GFP-ALLO-1 and GFP-IKKE-1 gradually increased until at least 20 min after sperm–oocyte contact in time-lapse imaging. Thus, the recognition of paternal mitochondria by ALLO-1 proteins appears to occur earlier than the recognition of MOs.

## ALLO-1 regulates recruitment of the ULK complex and IKKE-1

We previously reported that ALLO-1 directly binds to LGG-1, an Atg8/LC3 family protein in *C. elegans*, and regulates LGG-1 recruitment around paternal organelles[33]. However, whether ALLO-1 is involved in the recruitment of other autophagic factors remains unknown. We speculated that ALLO-1 directly regulates the recruitment of the ULK complex to initiate autophagosome formation around paternal organelles. UNC-51 and EPG-7/ATG-11 are homologs of mammalian ULK kinase and FIP200, the scaffold of the ULK complex, respectively[38,39] and are both required for allophagy[33]. In wild-type zygotes, EPG-7-GFP and UNC-51-GFP localized around the paternal organelles at the early 1-cell stage (around meiosis I, Supplementary Fig. 5a, b). Their localization appeared transient and became less clear after the pronuclear meeting stage (typically 30–50 min after fertilization). Live imaging showed that EPG-7-GFP localized earlier than LGG-1 within 1 min after sperm–oocyte contact, although the precise timing of its localization

could not be compared with that of ALLO-1 because of the relatively weak expression of EPG-7-GFP (Fig. 3e; Supplementary Movie 1). We observed that localization of EPG-7-GFP and UNC-51-GFP around paternal organelles was severely impaired in the *allo-1* mutant (Supplementary Fig. 5a, b), indicating that ALLO-1 is required for recruitment of the ULK complex around cargo.

As the N-terminal region of ALLO-1 binds tightly to IKKE-1[33], we further tested whether ALLO-1 regulates the localization of IKKE-1. In the *allo-1* mutant, GFP-IKKE-1 was not detected around paternal organelles (Supplementary Fig. 5c, d). These observations demonstrate that ALLO-1 regulates recruitment of the ULK complex and IKKE-1 around the cargo and is presumed to be the most upstream factor known in allophagy.

## IKKE-1 kinase activity controls ALLO-1b accumulation

Although *ikke-1* mutants are defective in allophagy, the precise role of IKKE-1 is unclear. We examined whether the *ikke-1* mutation affected ALLO-1 localization around paternal organelles. We first focused on ALLO-1b, which predominantly localizes to paternal mitochondria. GFP-ALLO-1b was observed around the paternal mitochondria, even in the *ikke-1* deletion mutant; however, the fluorescence intensity of GFP-ALLO-1b was significantly weaker than that of the wild type (Fig. 4a, b). A partial reduction in ALLO-1b localization around paternal

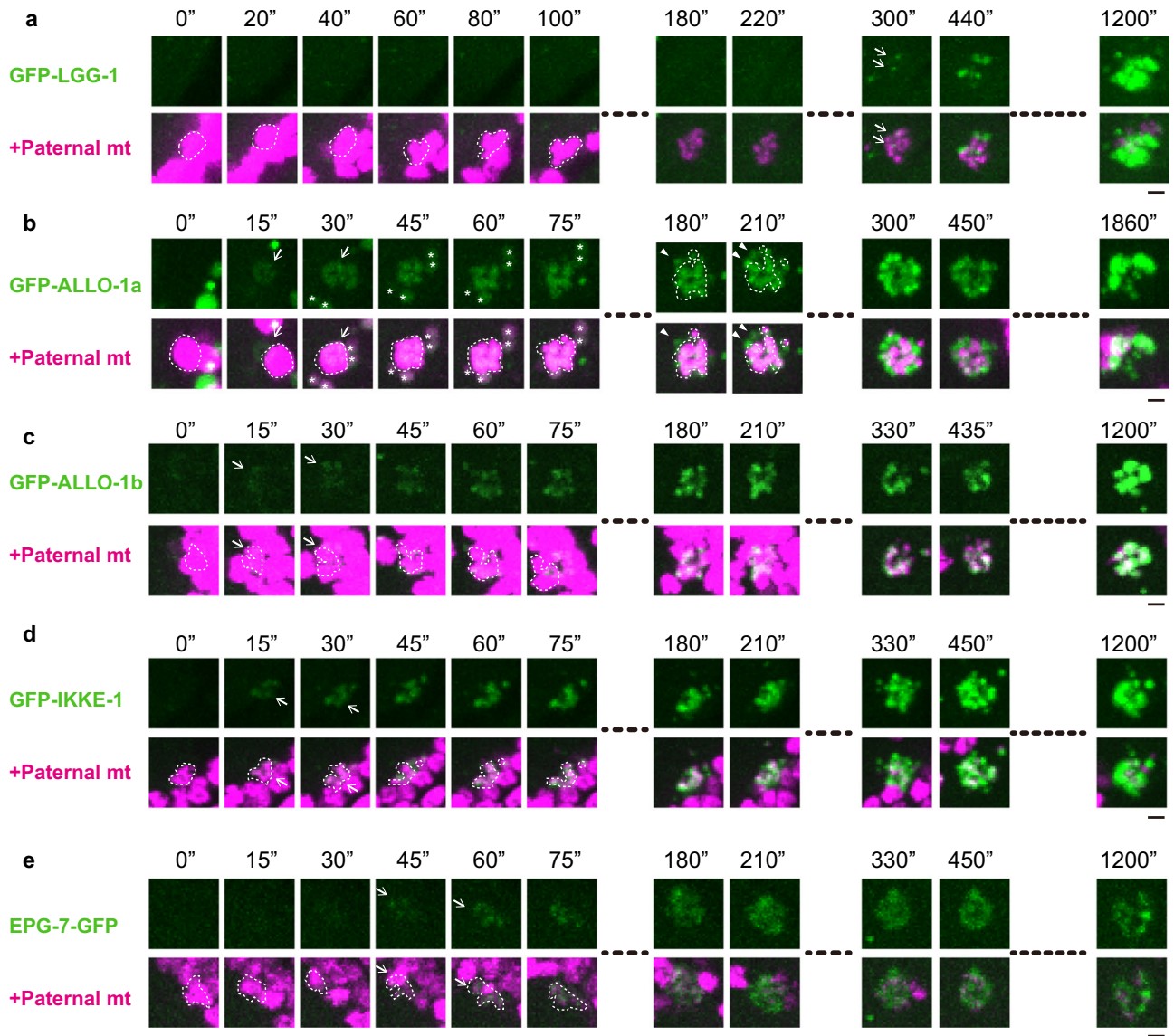

**Fig. 3 | Time-lapse imaging of allophagy factors. a–e** Time-lapse images of GFP-LGG-1 (**a**), GFP-ALLO-1a (**b**), GFP-ALLO-1b (**c**), GFP-IKKE-1 (**d**), and EPG-7-GFP (**e**) (green) along with HSP-6-mCherry (paternal mitochondria (mt); magenta) in zygotes. GFP-ALLO-1a or b was observed in the *allo-1* mutant background. The timing of sperm–oocyte contact was set to 0 s. Paternal mitochondria are surrounded by dotted white lines. Arrows indicate GFP localized to the paternal mitochondria. The maximum intensity projection images are shown. In **b**, the structures outside the paternal mitochondria, which were estimated as membranous organelles (MOs), are indicated by arrowheads. Asterisks indicate gut granules on the outside of the egg. Scale bars, 2 μm. See Supplementary Movie 1.

mitochondria was also observed in *ikke-1* kinase-dead mutants, suggesting the importance of kinase activity (Fig. 4a, b). Time-lapse imaging of GFP-ALLO-1b further revealed that the timing of the onset of ALLO-1b localization to paternal mitochondria after sperm–oocyte contact was unchanged in the *ikke-1* mutant. However, further accumulation of GFP-ALLO-1b molecules was impaired in the *ikke-1* mutant, resulting in a large difference in the amount of ALLO-1b around the paternal mitochondria between the wild type and *ikke-1* mutants at 20 min after sperm–oocyte contact (Fig. 4c; Supplementary Movie 2). We confirmed that the protein level of GFP-ALLO-1b was not affected by the *ikke-1* mutation (Supplementary Fig. 6a). Based on these observations, we hypothesized that localization of ALLO-1b is divided into at least two steps, as shown in Fig. 4d. In the first step, ALLO-1b recognizes the cargo and begins to localize within 30 s of sperm–oocyte contact, independently of IKKE-1 (cargo recognition). In the subsequent step, more ALLO-1b molecules accumulate around the cargo (accumulation) in a manner dependent on IKKE-1 kinase activity (Fig. 4a–c). Consistent with this model, a GFP-tagged C-terminal

fragment of ALLO-1b (181-402), which lacks IKKE-1 binding ability[33], weakly localized to the paternal mitochondria but showed minimal further accumulation (Supplementary Fig. 6b–d). Furthermore, localization of endogenous ALLO-1b was partially impaired in *ikke-1*-deficient zygotes according to immunostaining with an ALLO-1b-specific antibody, suggesting that the accumulation step was impaired (Supplementary Fig. 6e, f).

## IKKE-1 is essential for autophagosome formation

We further examined the effect of the *ikke-1* mutation on the recruitment of other autophagy factors. Recruitment of EPG-7-GFP was impaired in the *allo-1* mutant (Supplementary Fig. 5a; Fig. 5b). In contrast, in the *ikke-1* mutant, EPG-7-GFP was detected on paternal mitochondria but its fluorescence intensity was significantly reduced (Fig. 5a, b; Supplementary Fig. 6g). Thus, IKKE-1 is required for efficient accumulation of both ALLO-1 and EPG-7 on paternal mitochondria. We previously showed that GFP-LGG-1 recruitment around the paternal organelles is severely impaired in both *allo-1* and *ikke-1* mutants,

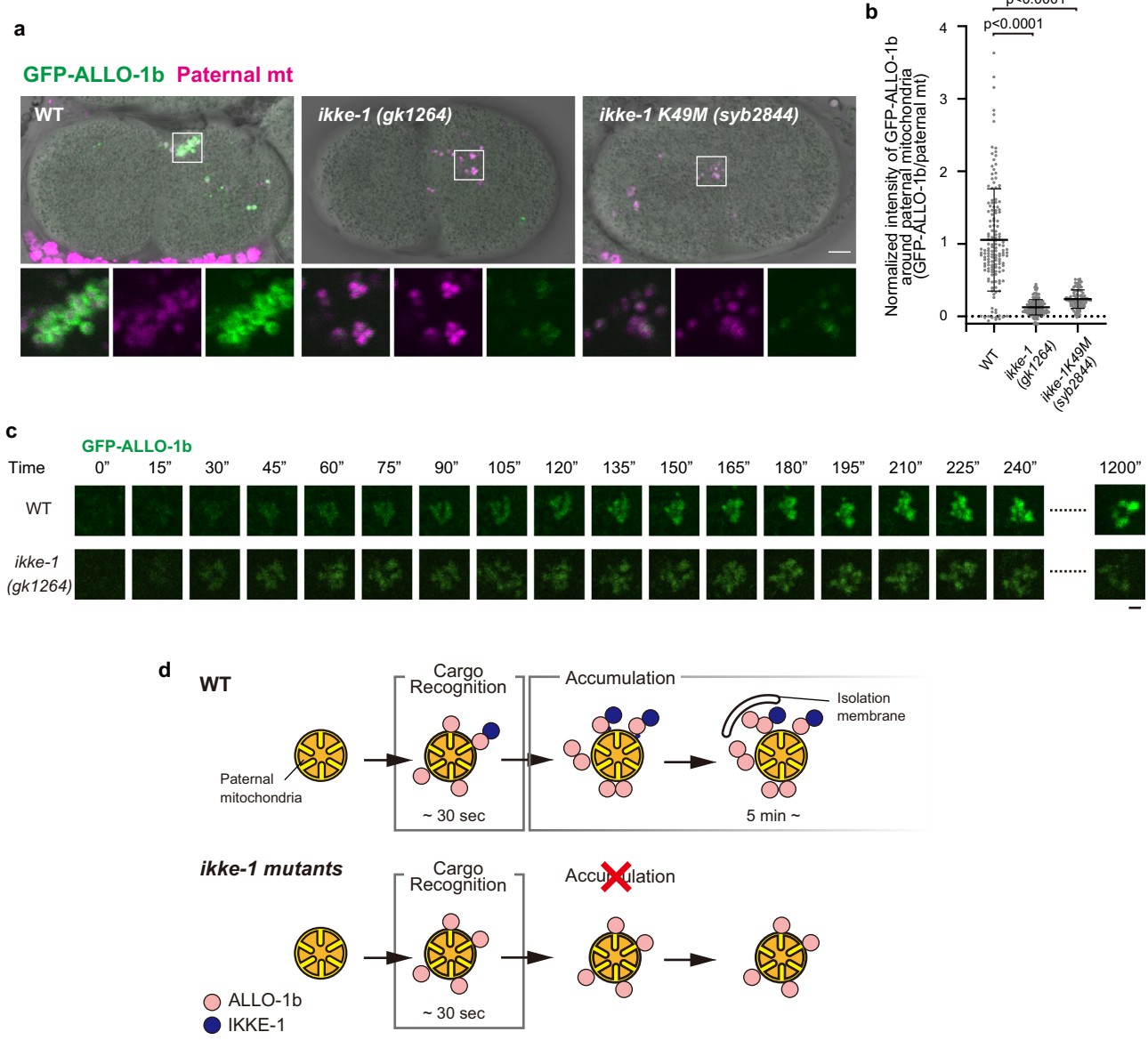

**Fig. 4 | IKKE-1 is essential for ALLO-1b accumulation. a, b** Accumulation of GFP-ALLO-1b was impaired in *ikke-1* deletion or kinase-dead (K49M) mutants. Zygotes (pseudo-cleavage stage) dissected from adult hermaphrodites expressing GFP-ALLO-1b (green) and HSP-6-mCherry (paternal mitochondria (mt); magenta) in the wild type (WT) or *ikke-1* mutant background were observed (**a**). Intensity of GFP-ALLO-1b was quantified and normalized using fluorescent HSP-6-mCherry (paternal mitochondria) to correct for GFP attenuation because of the distance from the objective (**b**). *n* = 150 (wild type, WT), *n* = 151 (*ikke-1(gk1264)*), and *n* = 96 (*ikke-1K49M(syb2844)*) paternal mitochondria or their clusters from 10 zygotes (pseudo-cleavage stage) for each strain were analyzed in **b**. Error bars represent the mean ± standard deviation (SD). Source data are provided as a Source Data file. Kruskal–Wallis with Steel-Dwass pairwise comparison test was performed for *p* values, which are provided in the figures and source data file. Scale bar, 5 µm. **c** Time-lapse images of GFP-ALLO-1b in the wild type and *ikke-1* deletion mutant zygotes. The timing of sperm–oocyte contact was set to 0 s. Scale bar, 2 µm. See Supplementary Movie 2. **d** Schematic representation of the relationship between IKKE-1 and ALLO-1b.

although the *ikke-1* mutant shows a slightly milder phenotype[33]. We verified the detailed GFP-LGG-1 localization pattern. In the wild-type zygotes, paternal mitochondria were surrounded by a bright GFP-LGG-1 signal by the pronuclear meeting stage (typically approximately 30–50 min after fertilization; Fig. 1a; Fig. 5c, d). We performed 3D reconstruction of these structures using super-resolution microscopy and observed that GFP-LGG-1 enclosed the paternal mitochondria (Fig. 5f; Supplementary Movie 3). In contrast, the association of the GFP-LGG-1 signal with paternal mitochondria was rarely observed in the *allo-1* mutant at the same stage, confirming failure of the initiation of autophagosome formation (Fig. 5c, d). In the *ikke-1* deletion mutants (two different alleles were used), the GFP-LGG-1 signal was undetectable in approximately 68–77% of paternal mitochondria observed,

suggesting that the partial accumulation of ALLO-1 and EPG-7 in the *ikke-1* mutant was insufficient to initiate LGG-1-positive isolation membrane formation. However, localization of small GFP-LGG-1 puncta was occasionally observed in the remaining 23–32% of paternal mitochondria in these *ikke-1* mutants. This partial localization of GFP-LGG-1 occurred in the kinase-dead mutant of *ikke-1* (Fig. 5c, d). 3D reconstruction analysis showed that the GFP-LGG-1-positive isolation membrane had only partially formed around the paternal mitochondria in the *ikke-1* mutant (Fig. 5f; Supplementary Movie 3). Consistent with these results, the total fluorescence intensity of GFP-LGG-1 around paternal mitochondria was significantly reduced in the *ikke-1* mutant (Fig. 5e). These results indicate that formation of LGG-1-positive isolation membranes was initiated on certain paternal mitochondria but

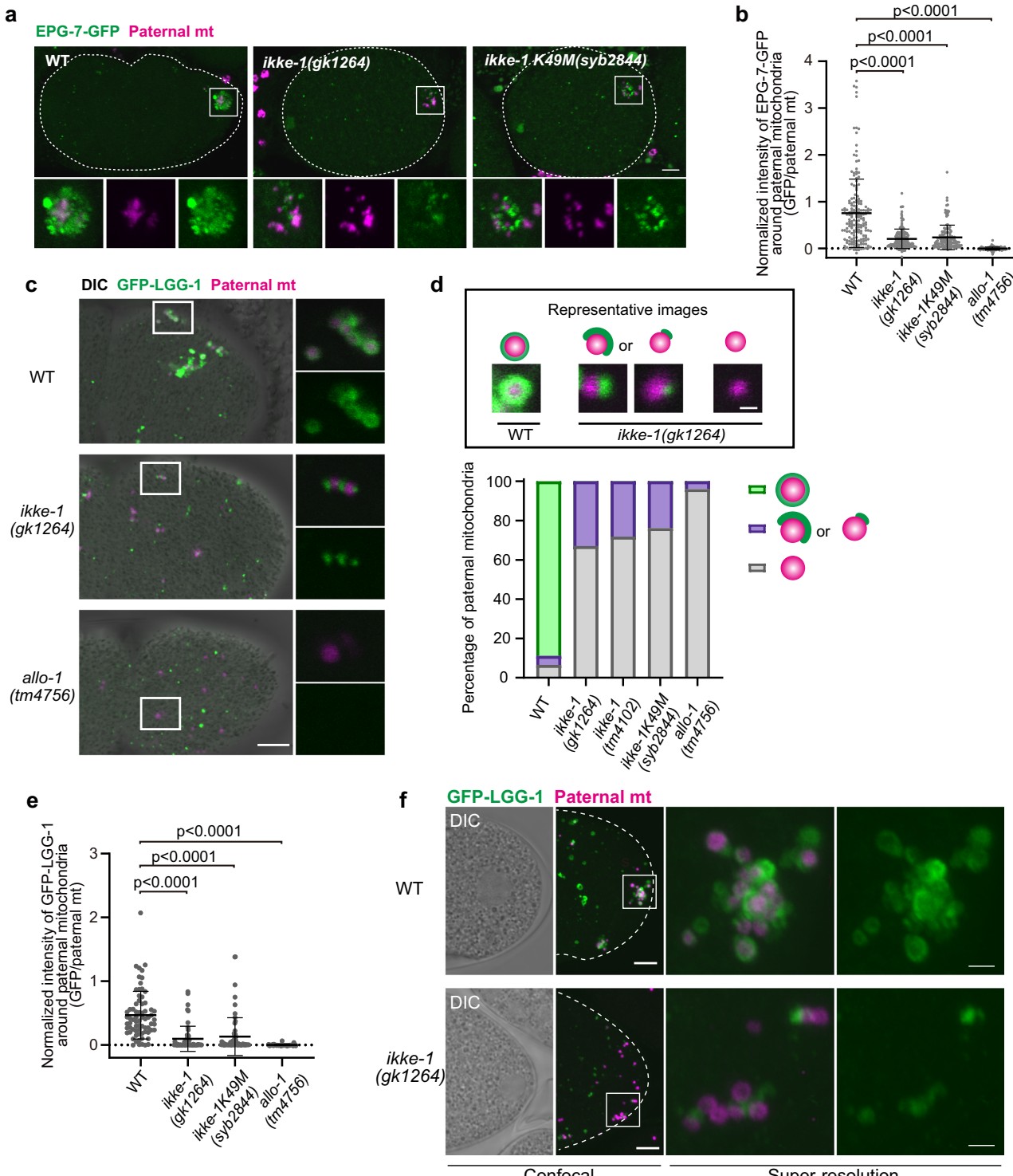

further membrane expansion was impaired without IKKE-1 kinase activity.

We investigated whether IKKE-1 functions similarly in MO degradation. ALLO-1a localized to MOs, in addition to paternal mitochondria, and functioned in MO degradation (Figs. 1c, d, g, h, 2 and 3b). The fluorescence intensity of GFP-ALLO-1a around the MOs was significantly reduced in the *ikke-1* mutant, although this reduction was not as pronounced as that of ALLO-1b around the paternal mitochondria (Fig. 6a, b). A pattern of GFP-LGG-1 localization around MOs was observed in the *ikke-1* mutant. Correlating with the mild effect on ALLO-1a accumulation, we found that more than half of the MOs

observed in the *ikke-1* mutants were partially surrounded by GFP-LGG-1 (Fig. 6c, d). The fluorescence intensity of GFP-LGG-1 around the MOs was weaker than that in the wild type, confirming partial LGG-1 recruitment (Fig. 6e). The GFP-LGG-1 signal was not detected in approximately 40% of MOs observed in *ikke-1* mutants (Fig. 6c, d). These data show that IKKE-1 is required for ALLO-1a accumulation and for the initiation/progression of autophagosome formation around MOs, but the requirement for IKKE-1 is relatively low compared to that for paternal mitochondrial degradation. It is also suggested that the level of ALLO-1 accumulation around the cargo is directly correlated with the level of GFP-LGG-1 recruitment around paternal organelles.

**Fig. 5 | IKKE-1 kinase activity is essential for autophagosome formation and EPG-7 accumulation around paternal mitochondria. a** EPG-7-GFP (green) and HSP-6-mCherry (paternal mitochondria (mt); magenta) were observed in the wild-type (WT) or *ikke-1* mutant zygotes (meiosis II stage). Scale bar, 5 μm. **b** Intensity of EPG-7-GFP was quantified and normalized using fluorescent HSP-6-mCherry. *n* = 177 (WT), *n* = 181 (*ikke-1(gk1264)*), *n* = 137 (*ikke-1K49M(syb2844)*), and *n* = 198 (*allo-1(tm4756)*) paternal mitochondria or their clusters from 10 zygotes (meiosis II) for each genotype. **c** GFP-LGG-1 (green) and HSP-6-mCherry (magenta) were observed in the pseudo-cleavage stage zygotes in the WT, *ikke-1* or *allo-1* mutant background. Scale bar, 5 μm. **d** The shape of GFP-LGG-1 surrounding paternal mitochondrion was observed in the zygotes (between the pronuclear expansion to the pseudo-cleavage stage) and classified into three types: not at all surrounded (gray), partially surrounded (purple), and almost completely surrounded (green). Representative images (single plane) of each shape are shown in the upper panel (scale bar, 500 nm). *n* = 216 (WT), *n* = 179 (*ikke-1(gk1264)*), *n* = 164 (*ikke-1K49M(syb2844)*),

*n* = 188 (*ikke-1(tm4102)*), and *n* = 203 (*allo-1(tm4756)*) paternal mitochondria from 15 zygotes for each genotype. **e** Normalized intensity of GFP-LGG-1 around paternal mitochondria. *n* = 71 (WT), *n* = 58 (*ikke-1(gk1264)*), *n* = 63 (*ikke-1K49M(syb2844)*), and *n* = 90 (*allo-1(tm4756)*) paternal mitochondria or their clusters from 10, 10, 10, or 11 zygotes (between the pronuclear expansion to the pseudo-cleavage stage), respectively. Scale bars, 5 μm. **f** Super-resolution images of GFP-LGG-1 (green) and paternal mitochondria (magenta) in the WT or *ikke-1(gk1264)* zygotes at the pronuclear expansion stage. The fluorescent images (z-projection) and differential interference contrast (DIC) images (single plane) are shown. Scale bars, 5 μm for confocal images and 1 μm for super-resolution images. See Supplementary Movie 3. White dotted lines indicate outline of the zygotes. Error bars represent the mean ± standard deviation (SD). Source data are provided as a Source Data file. Kruskal–Wallis with Steel-Dwass pairwise comparison test was performed for *p* values, which are provided in the figures and source data file.

## ALLO-1 directly binds to EPG-7

To investigate the mechanisms underlying ALLO-1 accumulation, we focused on the relationship between ALLO-1 and the ULK initiation complex. Several mammalian autophagy adaptors/receptors directly interact with FIP200/ATG11[21–24,40]. Analysis of whether ALLO-1 directly binds to EPG-7 using a yeast two-hybrid system showed that ALLO-1 interacted with the C-terminal fragment of EPG-7 (residues 1227–1338), which corresponds to the Claw domain of FIP200, similar to mammalian autophagy adaptors (Fig. 7a; Supplementary Fig. 7a)[21]. Additionally, the N-terminal region of ALLO-1 (residues 1–180) bound to EPG-7 (Fig. 7a). This N-terminal region is common to ALLO-1 isoforms and contains an LIR motif[33]. Several autophagy adaptors bind to FIP200 via FIP200-interacting region (FIR) motifs, which overlap with LIR motifs[21]. This FIR motif was well-conserved in ALLO-1 (Fig. 7b). When the putative FIR of ALLO-1 was mutated (D11A/E12A, D11A, or E12A), the interaction between ALLO-1 and the C-terminal fragment of EPG-7 was strongly impaired. However, these mutants still interacted with LGG-1 (Fig. 7c, d). Mutations in the LIR motif of ALLO-1 (F13A/I16A) impair its interaction with EPG-7, suggesting that these residues are important for both EPG-7 and LGG-1 binding, similar to other autophagy adaptors[21]. We confirmed that ALLO-1 directly bound to EPG-7 via this FIR motif in an in vitro binding assay using proteins synthesized in a cell-free system (Fig. 7e; Supplementary Fig. 7e). When the ALLO-1 FIR mutant was expressed in the *allo-1* mutant, allophagy defects were not efficiently rescued (Supplementary Fig. 7b–d). Based on these results, ALLO-1 directly binds to EPG-7 via the FIR motif, and this interaction is important for allophagy.

## EPG-7 is involved in ALLO-1 accumulation

We further investigated whether the *epg-7* mutation affects ALLO-1 accumulation. In an *epg-7* deletion mutant, GFP-ALLO-1b was detected around the paternal mitochondria but its intensity was significantly lower than that of the wild type and comparable to that of *ikke-1*-deficient and *ikke-1* kinase-dead mutants (Fig. 8a, b). Furthermore, the FIR mutant of ALLO-1b, which was unable to bind EPG-7 (Fig. 7c, e), showed a partial defect in ALLO-1 accumulation (Fig. 8c, d). The expression levels of the GFP transgenes were comparable (Fig. 8e). These results suggest that ALLO-1 weakly localizes to paternal organelles without EPG-7 but the accumulation of ALLO-1 around cargos depends on the interaction between ALLO-1 and EPG-7 as well as IKKE-1-dependent phosphorylation.

## IKKE-1 is involved in phosphorylation of ATG components

Our results suggest that IKKE-1-mediated phosphorylation is required for ALLO-1 and EPG-7 accumulation around paternal organelles, which leads to the initiation of isolation membrane formation. To identify the phosphorylated substrates of IKKE-1 that regulate this early step in allophagy, we comprehensively quantified the phosphoproteins using mass spectrometry. Early embryos, including the 1–100-cell stages,

were isolated from hermaphrodites of the wild type and mutants of either *allo-1* or *ikke-1*, and tryptic phosphopeptides prepared from these embryos were analyzed using isobaric tandem mass tag (TMT) labeling combined with liquid chromatography-tandem mass spectrometry (Fig. 8f, g; Supplementary Tables 1 and 2; Supplementary Data 1). The levels of two phosphopeptides were greatly reduced in the *ikke-1* mutants. Both peptides contained phosphorylated S949 of EPG-7 (Fig. 8f, red arrows). Moreover, peptides containing phosphorylated S949 of EPG-7 were reduced in *allo-1*-deficient embryos (Fig. 8g, red arrows). These results imply that IKKE-1-dependent phosphorylation of EPG-7 could be involved in the accumulation of ALLO-1 and EPG-7. To test this hypothesis, we mutated the serine residue to alanine (S949A). However, this single S949A mutation hardly affected paternal mitochondrial degradation (Supplementary Fig. 8b and c), and the phosphomimetic S949D mutation was unable to bypass the IKKE-1 requirement (Supplementary Fig. 8d, e). Enzymatic digestion and mass spectrometry of EPG-7-GFP immunoprecipitated from wild-type or *ikke-1* mutant embryos revealed at least 13 phosphorylation sites on EPG-7 other than S949 (Supplementary Fig. 8a; Supplementary Table 3; Supplementary Data 2). These phosphorylation sites were simultaneously mutated to alanine, but no obvious defect in allophagy was observed (Supplementary Fig. 8b, c). EPG-7 is a large protein composed of 1338 amino acids, and additional unidentified phosphorylation sites may be involved in this process. In addition to EPG-7, TMT analysis identified phosphopeptides of ALLO-1 and several autophagy regulators that control the early steps of autophagy, including the ULK1/2 homolog UNC-51, the ATG13 homolog EPG-1, and ATG-2, which were partially downregulated in the *ikke-1* mutant (Supplementary Table 1). Therefore, phosphorylation of multiple autophagy regulators other than EPG-7 by IKKE-1 may be involved in ALLO-1 accumulation (Fig. 8h).

## Discussion

Allophagy is a rapid process that occurs synchronously at every fertilization stage without artificial treatment. In addition, the entire process can be monitored in living animals and thus is a good model system for studying the initiation mechanism of selective autophagy around cargo in vivo. We observed rapid ALLO-1 localization to paternal organelles immediately after fertilization. In addition, recruitment of the UNC-51 initiation complex and IKKE-1 depended on ALLO-1, indicating that ALLO-1 is the most upstream regulator of allophagy among known factors. Detailed phenotypic analysis further revealed the stepwise recruitment of factors that initiate autophagy and the involvement of IKKE-1 kinase activity in this process (Supplementary Fig. 9). We propose a model in which the early stages of autophagy can be mechanistically divided into at least two steps based on ALLO-1 localization (Fig. 4d). First, ALLO-1 was weakly localized around paternal organelles immediately after fertilization (cargo recognition). This step is independent of IKKE-1 and EPG-7 and is

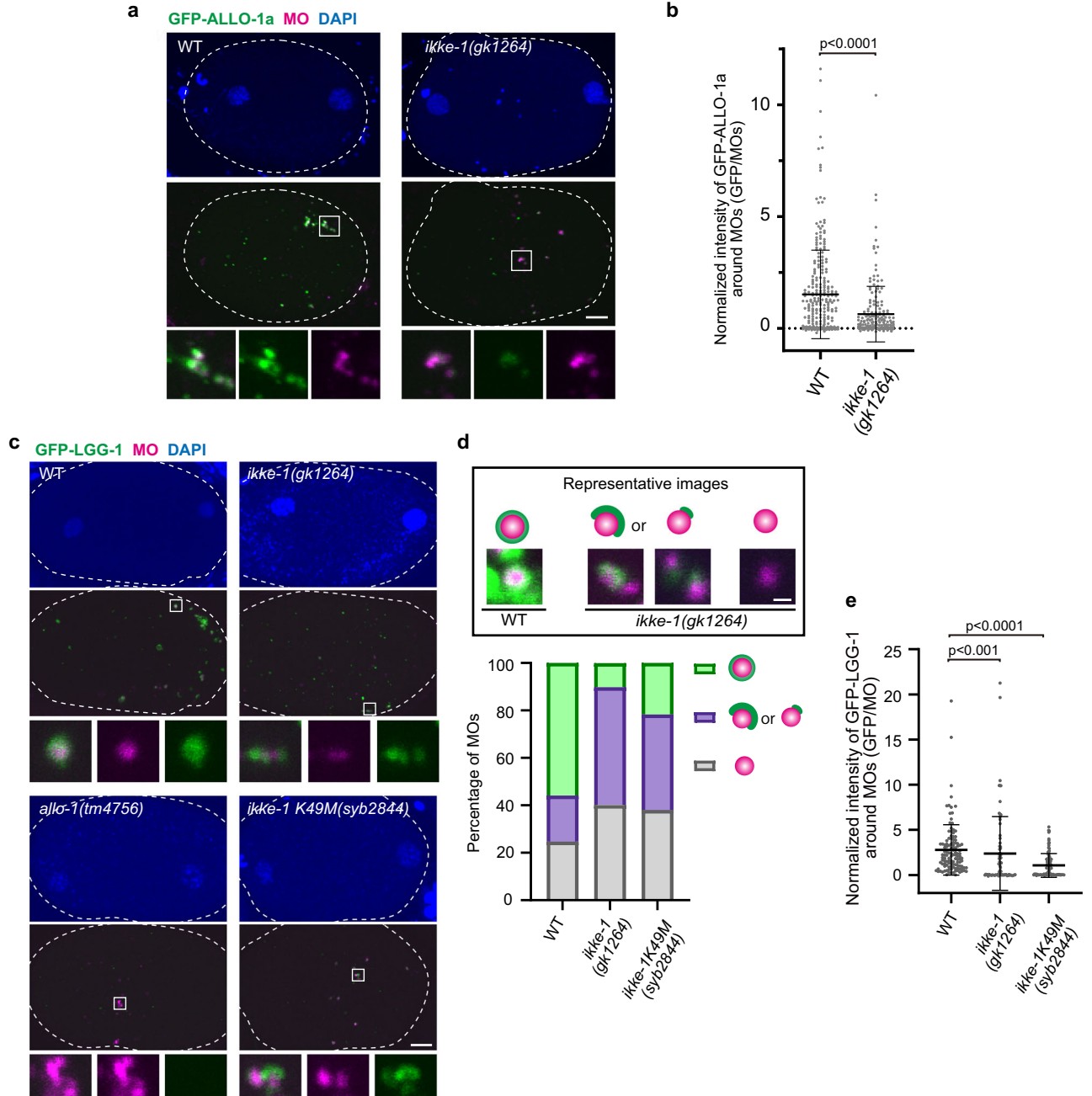

**Fig. 6 | IKKE-1 is essential for autophagosome formation and ALLO-1a accumulation around membranous organelles (MOs). a** IKKE-1 is necessary for ALLO-1a accumulation around MOs. Pseudo-cleavage stage zygotes were dissected from adult hermaphrodites expressing GFP-ALLO-1a in the *allo-1* mutant background and stained with an anti-MO antibody (1CB4) and 4′,6-diamidino-2-phenylindole (DAPI). **b** Intensity of GFP-ALLO-1a was quantified and normalized based on the fluorescence of MOs. *n* = 274 (wild type, WT) and *n* = 190 (*ikke-1(gk1264)*) MOs or their clusters were analyzed from 10 zygotes (between the pronuclear expansion to the pseudo-cleavage stage) for each genotype. **c** IKKE-1 and ALLO-1 are necessary for precise autophagosome formation around paternal mitochondria. Zygotes expressing GFP-LGG-1 were dissected from wild type, *allo-1(tm4756)*, *ikke-1(gk1264)*, and *ikke-1K49M(syb2844)* and stained with an 1CB4 antibody and DAPI between the pronuclear expansion to the pseudo-cleavage stage. **d** IKKE-1 kinase activity is necessary for autophagosome formation around paternal mitochondria. Zygotes expressing GFP-LGG-1 were stained with an anti-MO antibody. The shape of

GFP-LGG-1 surrounding MOs was observed and classified into three types: not at all surrounded (gray), partially surrounded (purple), and almost completely surrounded (green). *n* = 154 MOs from 10 zygotes (WT), *n* = 147 (*ikke-1(gk1264)*) MOs from 10 zygotes, and *n* = 92 (*ikke-1K49M(syb2844)*) MOs from 11 zygotes between the pronuclear expansion to the pseudo-cleavage stage. Representative images (single plane) of each shape are shown in the upper panel (scale bar, 500 nm). **e** Normalized intensity of GFP-LGG-1 around MOs. *n* = 119 (WT), *n* = 77 (*ikke-1(gk1264)*), and *n* = 102 (*ikke-1K49M(syb2844)*) MOs or their clusters from 13, 11, or 11 zygotes (between the pronuclear expansion to the pseudo-cleavage stage), respectively, were analyzed. *p* values were calculated using the two-tailed Mann–Whitney *U* test (**b**) or Kruskal–Wallis with Steel-Dwass pairwise comparison test (**e**). Error bars represent the mean ± standard deviation (SD). Source data are provided as a Source Data file. White dotted lines indicate outline of the zygotes. Scale bars, 5 μm.

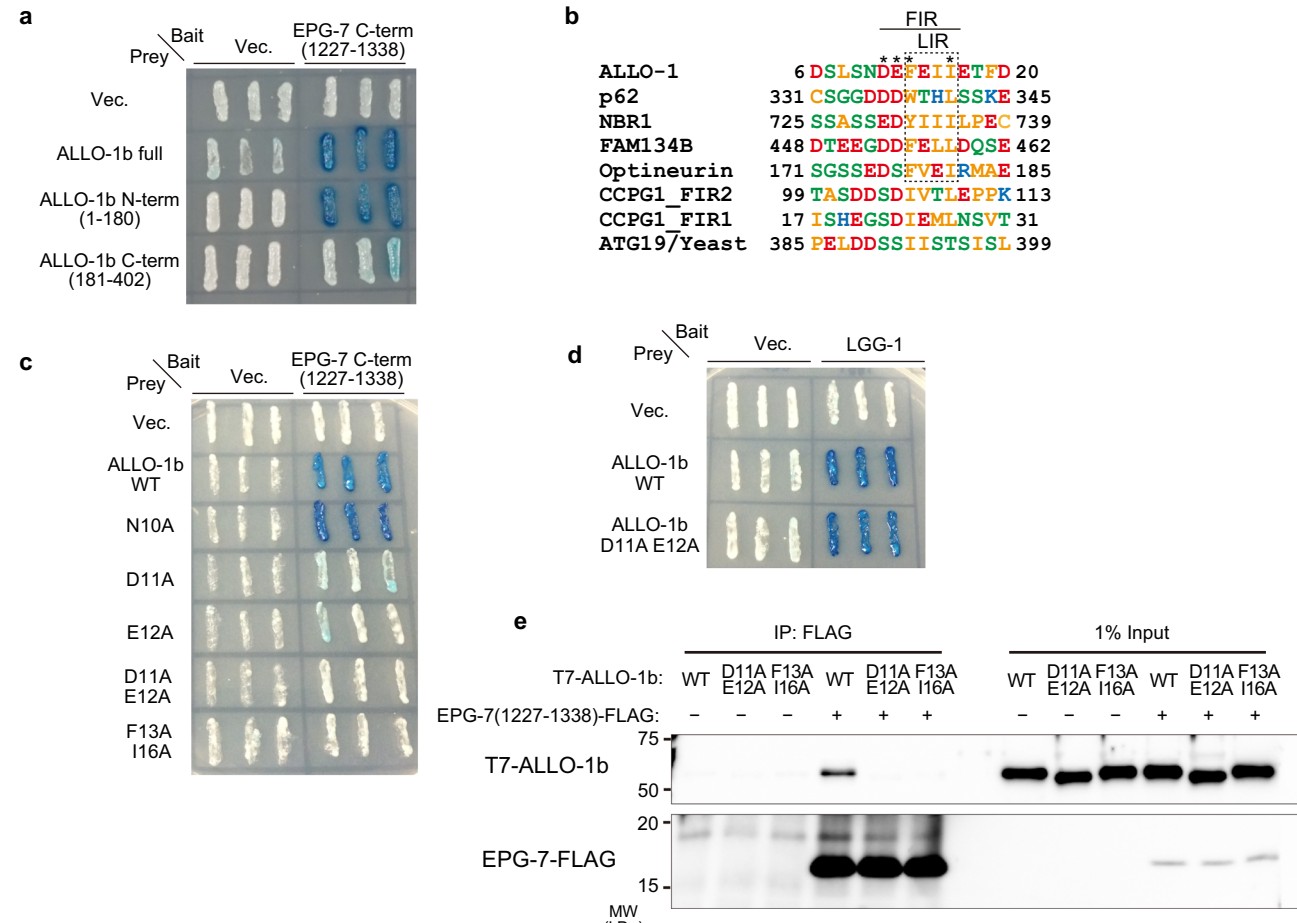

**Fig. 7 | ALLO-1 directly binds to EPG-7 via the FIP200-interacting region (FIR) motif. a** EPG-7 interacted with the ALLO-1 N-terminal region in yeast two-hybrid analysis. EPG-7 C-terminal region (1227–1338) corresponds to the Claw domain (see Supplementary Fig. 7a). **b** Sequence alignment analysis of the FIR or LIR motif of p62, NBR1, FAM134B, OPTN, and CCPG1 from human species and *Saccharomyces cerevisiae* ATG19. Highly conserved residues are marked with asterisks. **c** Interaction between EPG-7 and ALLO-1 depended on the FIR motif of ALLO-1 in yeast two-hybrid analysis. **d** FIR motif of ALLO-1 is not required for the interaction with LGG-1 according to yeast two-hybrid analysis. **e** In vitro binding assay confirmed the interaction between EPG-7 and ALLO-1 via the FIR motif. Unprocessed immunoblots are shown in Supplementary Fig. 7e. All experiments were independently repeated twice (**a**, **c**, **d**) or three times (**e**) with similar results.

mediated by the C-terminus of ALLO-1. Subsequently, ALLO-1 gradually accumulates around the cargo (accumulation). This accumulation step requires the kinase activity of IKKE-1, suggesting that a critical function of IKKE-1 is to facilitate the accumulation of ALLO-1, along with other autophagy proteins such as EPG-7, around the cargo.

As ALLO-1 accumulation proceeded, GFP-LGG-1 first appeared as puncta associated with the cargo approximately 5 min after sperm–oocyte contact and then gradually enclosed the cargo (Fig. 3a). EPG-7-GFP and GFP-LGG-1 were barely detected around the paternal organelles without ALLO-1 (Supplementary Fig. 5; Fig. 5c, d)[33]. Although ALLO-1 and EPG-7 were weakly localized in the *ikke-1* mutant; GFP-LGG-1 was not detected around most paternal organelles. This observation suggests that accumulation of ALLO-1 and EPG-7 above a certain threshold is required to initiate LGG-1-positive isolation membrane formation around the cargo. In contrast to the *allo-1* mutant, partial LGG-1 recruitment was observed in certain paternal organelles in the *ikke-1* mutant (Fig. 5c, d). Furthermore, a comparison of the allophagy of paternal mitochondria and MOs suggested that the frequency of appearance and size of LGG-1-positive isolation membranes correlated with the level of ALLO-1 accumulation. In addition to autophagy initiation, IKKE-1-mediated ALLO-1 accumulation may be required for membrane expansion around the cargo by facilitating recruitment of the ULK complex and downstream autophagy regulators. A recent study reported that TBK1 and ULK1/2 function

redundantly in mammalian NDP52-mediated mitophagy, and mitophagy proceeds with only one of these kinases[32]. In contrast, both IKKE-1 and UNC-51 are required for normal allophagy[6,33]. As allophagy is a rapid reaction compared to PINK1–Parkin-dependent mitophagy, the activation of both kinases may be necessary to drive such a reaction. IKKE-1 and UNC-51 may function additively or sequentially during allophagy.

ALLO-1 directly bound to the C-terminal Claw domain of EPG-7 via the FIR motif. ALLO-1 FIR motif mutants, which still bind to LGG-1 in the yeast two-hybrid assay, were unable to rescue the *allo-1* mutant, suggesting that interaction with EPG-7 is important for ALLO-1 function. Autophagy adaptors, such as NDP52 and p62, bind to FIP200 and mediate the recruitment of the ULK complex in mammalian cells[21,40]. Yeast autophagy receptors/adaptors bind to Atg11, a scaffold of the Atg1 complex, in selective autophagy pathways[41–44]. Therefore, Atg11/FIP200-binding is a widely conserved feature of autophagy receptors/adaptors, in addition to their ability to bind to Atg8/LC3. In mammals, OPTN and NDP52 function redundantly in PINK1-Parkin-dependent mitophagy and directly bind to TBK1 and FIP200, respectively[22,24,45]. Although OPTN and NDP52 are not conserved in *C. elegans*, ALLO-1 may have both OPTN- and NDP52-like functions.

The accumulation of ALLO-1 depends on the interaction between ALLO-1 and EPG-7, suggesting a positive feedback mechanism in which recruitment of EPG-7 facilitates further ALLO-1 recruitment or

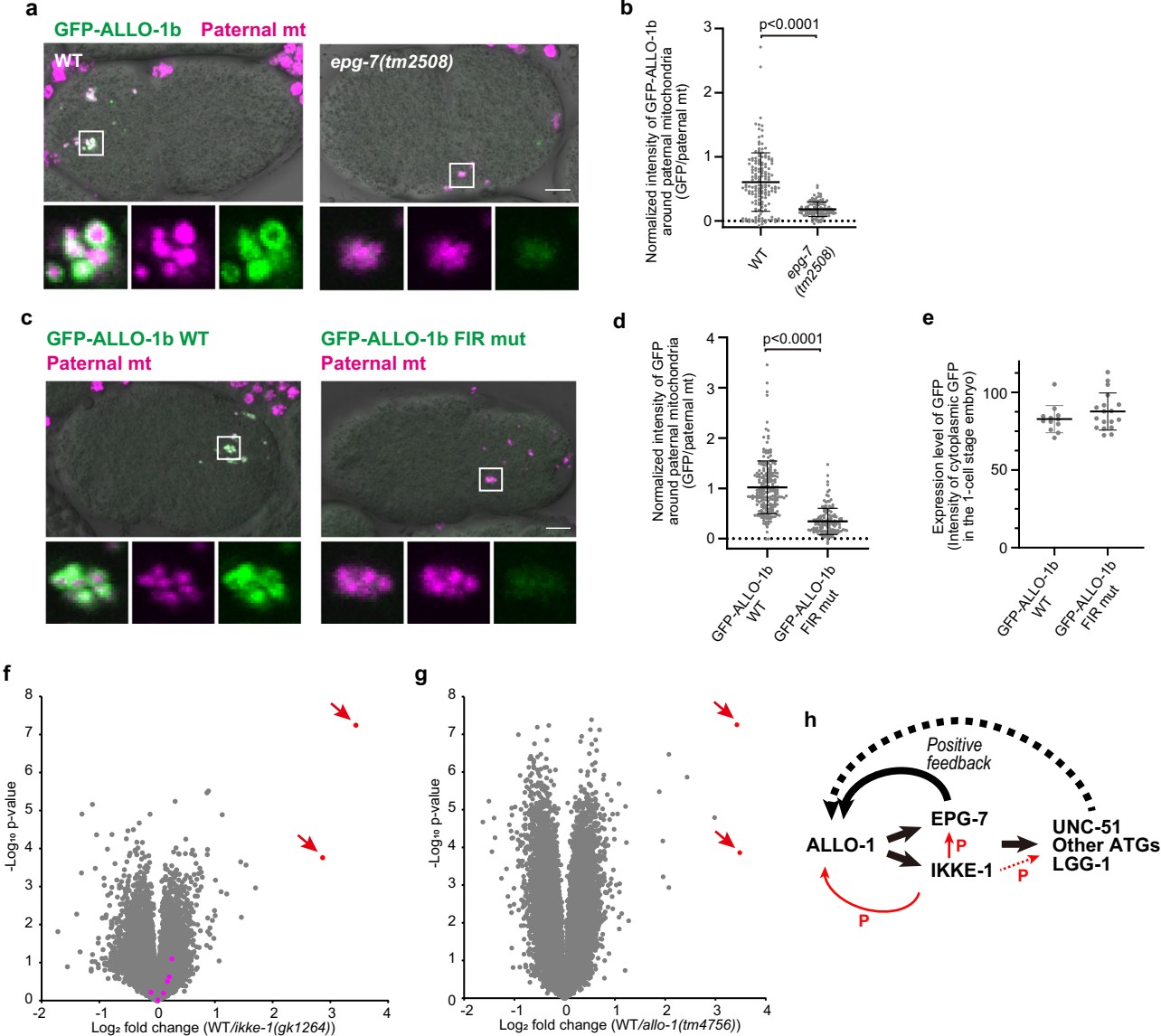

**Fig. 8 | EPG-7 is involved in ALLO-1b accumulation. a, b** EPG-7 is necessary for the accumulation of ALLO-1b around paternal mitochondria (mt). GFP-ALLO-1b (green) and HSP-6-mCherry (paternal mt; magenta) were observed in the wild-type (WT) or *epg-7* mutant zygotes. In **b**, the intensity of GFP-ALLO-1b was quantified and normalized based on the fluorescence of paternal mitochondria. *n* = 154 (wild type, WT) and *n* = 108 (*epg-7(tm2508)*) paternal mitochondria or their clusters from 9 or 10 zygotes, respectively, were analyzed. All zygotes were imaged between the pronuclear expansion to the pseudo-cleavage stage. **c–e** Accumulation of ALLO-1b depends on the FIP200-interacting region (FIR) motif. GFP-ALLO-1b WT or GFP-ALLO-1b D11AE12A (FIR mut) (green) and HSP-6-mCherry (paternal mt; magenta) were observed in zygotes. In **d**, the intensity of GFP co-localized with paternal mitochondria was quantified. *n* = 207 (GFP-ALLO-1b WT) and *n* = 140 (GFP-ALLO-1b FIR mut) paternal mitochondria or their clusters from 10 zygotes for each genotype were analyzed. In **e**, the intensity of cytoplasmic GFP was quantified outside the

vicinity of the paternal organelles to compare GFP expression levels between transgenes (*n* = 12 or 18 zygotes for WT or FIRmut, respectively). All zygotes were imaged between the pronuclear expansion to the pseudo-cleavage stage. Volcano plot of tandem mass tag (TMT)-based quantitative phosphoproteomics identifying the decrease in phosphorylation of EPG-7 S949 in *ikke-1* (**f**) and *allo-1* (**g**) mutants. Red arrows indicate phosphorylated S949-containing peptides of EPG-7. Magenta dots in **f** indicate the peptides of ALLO-1 (see Supplementary Table 2). The *x*-axis shows the ratio of wild type to the mutants in the amount of detected peptides. *p* values in the volcano plots were calculated by the two-tailed Student's *t*-test. **h** Model of ALLO-1 accumulation via a positive feedback loop by IKKE-1. P, phosphorylation. Scale bars, 5 μm. *p* values were calculated using the two-tailed Mann–Whitney *U* test (**b, d, e**). Error bars represent the mean ± standard deviation (SD). Source data are provided as a Source Data file.

stabilizes ALLO-1 localization, resulting in the accumulation of more ALLO-1 molecules around the cargo (Fig. 8h). Expansion of the isolation membrane may also contribute to the stabilization of ALLO-1 localization via the interaction between ALLO-1 and LGG-1 on the membrane, as suggested for OPTN and NDP52 in mitophagy in mammals and Atg40 in endoplasmic reticulum-phagy in yeast[46,47].

Because the kinase activity of IKKE-1 is indispensable for driving ALLO-1 accumulation, we examined its phosphorylation substrates. EPG-7 is a highly phosphorylated protein, and its phosphorylation level

of S949 was reduced in the *ikke-1* mutant. However, no obvious allophagy defects were observed in the phosphorylation site mutants. Therefore, it is possible that IKKE-1-dependent phosphorylation sites of EPG-7, which were not detected in this analysis, or IKKE-1-dependent phosphorylation of autophagy factors other than EPG-7 may enhance or stabilize the EPG-7-ALLO-1 interaction and contribute to ALLO-1 accumulation. We previously reported that ALLO-1 is phosphorylated in an IKKE-1-dependent manner. This phosphorylation is involved in efficient allophagy, but ALLO-1 mutants lacking IKKE-1-dependent

phosphorylation sites exhibited mild allophagy defects compared to those of *ikke-1* mutants[33]. Our comprehensive analysis suggests the presence of many other substrates of IKKE-1-dependent phosphorylation, including ATG-2, that are involved in the expansion of isolation membranes[48]. Additionally, EPG-7 forms an oligomer and binds to multiple autophagy regulators in various regions[38]. In addition to EPG-7 and ALLO-1, phosphorylation of multiple downstream proteins may synergistically enhance the accumulation of ALLO-1 and EPG-7 along with downstream autophagy factors around the cargo. The phosphorylation sites of EPG-7 are mapped to the oligomerization domain or regions that interact with multiple autophagy regulators (Supplementary Fig. 8a). As TBK1 phosphorylates multiple proteins involved in selective autophagy[26–28,30–32], TBK1/IKKE-1 may regulate multiple steps of selective autophagy via phosphorylation of different substrates.

ALLO-1 has two isoforms with different localization patterns. ALLO-1a localized to both paternal organelles and preferentially accumulated around MOs. In contrast, ALLO-1b preferentially accumulated around the paternal mitochondria (Fig. 1). ALLO-1a is widely conserved in nematode species, whereas the ALLO-1b homolog was not observed in current gene prediction of WormBase. However, only a limited number of cDNA have been registered in these species, and further gene expression analyses are required. We observed that accumulation of these isoforms on the cargo had different IKKE-1 requirements. ALLO-1a accumulation and subsequent LGG-1 recruitment to MOs were less sensitive to *ikke-1* mutation, although IKKE-1 was still required for the complete engulfment of MOs. Because IKKE-1 dependence appears to be inversely proportional to the amount of ubiquitin on paternal organelles, ubiquitin may promote ALLO-1 accumulation to a certain extent, even in the absence of IKKE-1. Finally, the timing of ALLO-1 localization around paternal mitochondria was more rapid than that around the MOs. Localization of ALLO-1a to MOs began approximately 3 min after sperm–oocyte contact. Ubiquitination of MOs may begin after fertilization, after which ALLO-1a localization occurs. This observation is consistent with the fact that MOs are ubiquitinated after fertilization[7]. In contrast, the localization of ALLO-1b to paternal mitochondria occurs within 30 s after sperm–oocyte contact, implying that sperm mitochondria are modified even before fertilization to be readily recognized by ALLO-1b. In other species such as *Cryptococcus*, *Drosophila*, and Japanese rice fish (*Oryzias latipes*), mtDNA or mtDNA–protein complexes (generally named as mitochondrial nucleoids) are eliminated or reduced before fertilization[12,13,49], indicating that paternal mitochondria undergo some changes before fertilization in these species. Unknown changes may occur in the mitochondria and mtDNA during spermatogenesis in *C. elegans* which are recognized by ALLO-1 after fertilization. Ubiquitination after fertilization may enhance or stabilize ALLO-1 localization. Further analysis is necessary to determine the differences between paternal and maternal mitochondria and the mechanism by which ALLO-1 recognizes these mitochondria.

## Methods

### Worm strains and maintenance
The strains used in this study are listed in Supplementary Table 4. *Caenorhabditis elegans* was maintained as previously described[50]. Briefly, strains were placed on the nematode growth medium petri plates (3 g NaCl, 2.5 g peptone, 17 g agar, 1 mL of 1 M CaCl$_2$, 1 mL of 1 M MgSO$_4$, 5 mg cholesterol, 25 mL of 1 M KPO$_4$ buffer pH 6.0, and up to 1 L distilled water) with *Escherichia coli* strain OP50, and grown at 20 or 25 °C. All experiments were performed after the nematodes had been incubated at 20 °C for at least 24 h. *ikke-1(syb2844)* mutant and *unc-51(syb2940)* strains were generated by SUNY Biotech using clustered regularly interspaced palindromic repeat (CRISPR)/CRISPR-associated protein 9 genome editing. In *ikke-1(syb2844)*, the codon corresponding to K49 was changed to ATG along with two additional silent mutations (GTGAAAACGGCT > GTCATGACGGCA). For *unc-51(syb2940)*, a

fragment encoding GFP was inserted immediately before the stop codon of *unc-51*. *ikke-1(gk1264)*, *him-5(e1490)* and *unc-119(ed3)* mutant strains were obtained from the Caenorhabditis Genetic Center supported by the NIH Office of Research Infrastructure Programs (P40 OD010440). The deletion mutants, *allo-1(tm4756)*, *ikke-1(tm4102)*, and *epg-7(tm2508)*, were provided by S. Mitani of the Japanese National Bioresource Project for the Experimental Animal "Nematode *C. elegans*." Each mutant was backcrossed with the wild-type N2 at least twice.

### Plasmids and transgenic *C. elegans*
To express one of the isoforms of ALLO-1 in *C. elegans*, the DNA sequence corresponding to residues 1–354 was cloned from the genome, and the sequence corresponding to the subsequent amino acids was cloned from cDNA. These two fragments were combined using polymerase chain reaction (PCR) and cloned into pDONR221 via Gateway Recombination using BP clonase (Thermo Fisher Scientific, Waltham, MA, USA). The fragments were subcloned into pID3.01B[51] to express GFP-tagged fusions under the *pie-1* promoter via Gateway Recombination, using LR clonase (Thermo Fisher Scientific). To express EPG-7-GFP, a genomic fragment containing *epg-7* and the GFP cording fragment derived from pID3.01B was amplified using PCR. A fragment of the vector containing *pie-1* promoter and terminator was amplified using PCR, using pID2.02[52] as the template. Fragments of *epg-7* and GFP were inserted into the pID2.02 vector using an In-Fusion Cloning Kit (Clontech, Mountain View, CA, USA). Mutations (D11A/E12A, S949A, and S949D) were generated using site-directed mutagenesis (New England Biolabs, Ipswich, MA, USA). To express ALLO-1a (355–388) or ALLO-1b (355–402), the DNA sequences corresponding to these amino acids were amplified using PCR from pDONR221 ALLO-1a or ALLO-1b, respectively, and cloned into pDONR221 using an In-Fusion Cloning Kit. The fragments were subcloned into pID3.01B to express GFP-tagged fusions under the *pie-1* promoter via Gateway Recombination, using LR clonase (Thermo Fisher Scientific).

To express EPG-7, which was mutated at all detected phosphorylation sites to alanine, the mutated DNA fragment was synthesized by Twist Bioscience (South San Francisco, CA, USA). The position of mutated sites in the protein, which includes possible phosphorylation sites, are as follows (epg-7S/T → A fragment); S188, T190, S209, S217, T220, S221, T222, S223, T224, S548, S578, S588, S600, S606, S620, S621, S625, S629, T630, S631, S632, S639, S644, S645, S653, S656, S657, T659, S660, T666, T742, S949, and S1204. The vector fragment containing *pie-1* promoter, terminator, and the GFP cording region was amplified using PCR with the pID2.02-epg-7-gfp vector prepared as described above. The synthesized epg-7S/T → A fragment was inserted in the vector fragment using an In-Fusion Cloning Kit. The primers used in this study are listed in Supplementary Table 5.

Transgenic worms were generated using the microparticle bombardment method, as previously described[53]. The following transgenic lines were used: *dkIs890[Ppie-1::gfp::allo-1a, unc-119(+)]*, *dkIs851[Ppie-1::gfp::allo-1b, unc-119(+)]*, *dkIs926[Ppie-1::epg-7::gfp, unc-119(+)]*, *dkIs985[Ppie-1::gfp::allo-1a + b D11A/E12A]*, *dkIs1011[Ppie-1::gfp::allo-1b D11A/E12A]*, *dkIs928[Ppie-1::epg-7S949A::gfp, unc-119(+)]*, *dkIs1015[Ppie-1::epg-7 S/T → A::gfp]*, *dkIs1128[Ppie-1::epg-7S949D::gfp, unc-119(+)]*, *dkIs1122[Ppie-1::gfp::allo-1a(355–388), unc-119(+)]*, *dkIs1124[Ppie-1::gfp::allo-1b(355–402), unc-119(+)]* (this study): *dkIs698[Pspe-11::hsp-6::mCherry, unc-119(+)]*, *dkIs737[Ppie-1::gfp::ikke-1, unc-119(+)]*, *dkIs811[Ppie-1::gfp::allo-1a + b, unc-119(+)]*, *dkIs836[Ppie-1::gfp::allo-1b C-term, unc-119(+)]*, *dkIs821[Ppie-1::gfp:: allo-1a + b F13A/I16A, unc-119(+)]*:[33] *dkIs623[Pspe-11::hsp-6::gfp, unc-119 (+)]*, and *dkIs398[Ppie-1::gfp::lgg-1, unc-119(+)]*[6].

### Yeast two-hybrid analysis
The full-length cDNA of *allo-1b*, C-terminal fragments of *allo-1b*, Claw domain of *epg-7*, and N-terminal fragment of *allo-1* were amplified via

PCR using the cDNA library of the DupLEX-A Yeast Two-Hybrid System (OriGene Technologies, Rockville, MD, USA) and cloned into pDONR221 using gateway recombination. These genes were subcloned into the bait plasmid pEG202Gtwy[54] and prey plasmid pJG4-5Gtwy[54] via gateway recombination. Mutations (F13A/I16A and D11A/E12A) were generated using site-directed mutagenesis. Plasmids were introduced into the yeast EGY48 strain harboring the lacZ reporter plasmid pSH18-34. The transformants were inoculated onto 5-bromo-4-chloro-3-indolyl β-D-galactopyranoside-containing plates.

## In vitro binding assay

The C-terminal sequence of EPG-7, corresponding to the FIP200 Claw domain, was determined using Clustal Omega[55]. The cDNA of the C-terminal fragment of *epg-7* was cloned into pT7-FLAG-1 (Sigma-Aldrich, St. Louis, MO, USA). The N-terminal FLAG-tagged EPG-7 C-term was expressed in *E. coli* Rosetta (DE3) cells. *E. coli* pellets were suspended in phosphate-buffered saline (PBS) containing 1 mM PMSF and 1% (v/v) TritonX-100 and sonicated on ice. The sonicated cells were centrifuged at $9100 \times g$ for 15 min at 4 °C, and the supernatant was incubated with 200 μL (slurry) of Anti-FLAG M2 Affinity Gel (Sigma-Aldrich). The N-terminal FLAG-tagged EPG-7 C-terminal peptides were eluted using 5 mg/mL FLAG peptide (Sigma-Aldrich).

Full-length ALLO-1b and its mutants (D11A/E12A and F13A/I16A) were synthesized in vitro using a cell-free protein synthesis system. Gene templates for cell-free protein synthesis were amplified using two-step PCR. In the first PCR step, genes were cloned from the template plasmid using specific primers for the first PCR (Supplementary Table 5). In the second PCR, primers 2nd_NF2, 2nd_T7_NF1, and 2nd_NCR_U01 were used to add the T7 promoter, enhancer, T7 tag, and linker sequence to the N-terminus of the first PCR product, and the tail sequence to the C-terminus (Supplementary Table 5). The resulting templates were transcribed and translated using a Cell-Free Protein Synthesis Kit (NU Protein, Tokushima, Japan).

Purified FLAG-EPG-7 C-term (approximately 0.5 μg) and in vitro translated T7-ALLO-1b, T7-ALLO-1b D11A/E12A, or T7-ALLO-1b F13A/I16A (4 μL) were incubated with 300 μL of co-immunoprecipitation buffer (10 mM Tris-HCl pH 7.4, 150 mM NaCl, 0.5 mM EDTA, 0.1% [v/v] TritonX-100, and 1 mM PMSF) at 4 °C for 1 h. Anti-FLAG M2 magnetic beads (Sigma-Aldrich) were added and further incubated at 4 °C for 1 h. The beads were washed with the same buffer and boiled with sodium dodecyl sulfate (SDS) sample buffer (62.5 mM Tris-HCl, pH 6.8, 2% [w/v] SDS, 10% [v/v] glycerol, 1% [v/v] 2-mercaptoethanol, and 0.01% [w/v] bromophenol blue) for elution. The bound proteins were analyzed using immunoblotting with an anti-T7-tag (PM022, Medical and Biological Laboratories, Tokyo, Japan, dilution 1:5000) and anti-FLAG 1E6 (018-22381, Fujifilm Wako Chemicals, Osaka, Japan, dilution 1:5000) antibodies and secondary antibodies shown in the next section.

## Immunoblotting

To prepare whole-worm lysates, 60 adult hermaphrodites were collected at 24 h after the L4 stage in M9 buffer (3 g $KH_2PO_4$, 6 g $Na_2HPO_4$, 5 g NaCl, 1 mL 1 M $MgSO_4$, and up to 1 L distilled water), washed twice with M9 buffer and boiled in SDS sample buffer for 10 min. To prepare fertilized-egg lysates, fertilized eggs were collected from adult hermaphrodites using the bleaching method, as previously described[56]. Briefly, the worms were collected using M9 buffer, sterilized by autoclaving, and mixed with an equal amount of bleaching solution (1 N NaOH, 0.5% [v/v] sodium hypochlorite). Fertilized eggs were collected by centrifugation after the nematode bodies were dissolved. Eggs were washed twice with M9 buffer, and boiled in SDS sample buffer for 10 min. The samples were then centrifuged and the supernatant was used. The samples were subjected to immunoblotting using an anti-GFP polyclonal antibody (70R-GG001, Fitzgerald Industries International, Acton, MA, USA, dilution 1:2000), an anti-actin antibody

(MAB1501, clone C4, Merck, Kenilworth, NJ, USA, dilution 1:5000), an anti-ALLO-1a antibody (this study, dilution 1:10,000), or an anti-ALLO-1b antibody (this study, dilution 1:1000) as primary antibodies. Peroxidase-conjugated donkey anti-goat immunoglobulin G antibody (AP180P, Merck), peroxidase-conjugated goat anti-mouse immunoglobulin G antibody (115-035-003, Jackson ImmunoResearch, West Grove, PA, USA), or peroxidase-conjugated goat anti-rabbit immunoglobulin G antibody (111-035-003, Jackson ImmunoResearch) were used as secondary antibodies at 1:5,000 dilution. SuperSignal™ West Pico PLUS Chemiluminescent Substrate (34580, Thermo Fisher Scientific) and SuperSignal™ West Dura (37071, Thermo Fisher Scientific) were used to visualize the signals, which were detected on a FUSION Solo 7 S (M&S Instruments, Osaka, Japan). Anti-ALLO-1a and ALLO-1b polyclonal antibodies were generated by TK Craft Corporation, Gunma, Japan against peptides corresponding to the ALLO-1a C-terminal region (QYFAKPTSSPSSSAK; Sigma-Aldrich) or ALLO-1b C-terminal region (NDPLDSLSNDEFEIIETFD; Sigma-Aldrich) and affinity-purified with the same peptides. The specificity of these antibodies was verified using immunoblotting and immunostaining of wild-type and *allo-1(tm4756)* strains.

## Microscopy

To obtain images of live zygotes and embryos, adult hermaphrodites were dissected to isolate the zygotes and embryos in M9 buffer with 20 mM levamisole and mounted on 1.5% agarose pads. Confocal images were obtained using an Olympus FV1200 confocal laser-scanning microscope consisting of an Olympus IX83 inverted microscope, 405-, 473-, 559-, and 635 nm LD lasers, a galvanometer mirror scanner, and a multi-alkali photomultiplier detector (Olympus, Tokyo, Japan). We used a 100 × 1.35 numerical aperture (NA) UPlanSApo or a 60 × 1.40 NA UPlanSApo oil-objective lens (Olympus) to obtain images.

To quantify the area of paternal mitochondria and MOs in the rescue experiments, projected images of confocal z-stacks covering the zygotes or embryos (1 μm intervals) were generated using FV10-ASW software ver.4.2b (Olympus); the images were binarized and the signal area inside the eggs was quantified using Fiji software[57]. To quantify the intensity of GFP-ALLO-1a, GFP-ALLO-1b, EPG-7-GFP, or GFP-IKKE-1 puncta, projected images of confocal z-stacks covering the zygotes or embryos (1 μm intervals) were generated using Fiji software. Note, however, that in analysis of the intensity and classification of GFP-LGG-1 in Figs. 5 and 6 and intensity of GFP-ALLO-1a around the paternal mitochondria in Supplementary Fig. 1, we analyzed a single z-slice image to avoid the influence of bright signals in close proximity. A region of interest (ROI) was manually created around the paternal mitochondria or MOs, and the mean intensity of GFP and paternal mitochondria or MOs in the ROI was measured using Fiji software. When MOs or mitochondria formed clusters that were difficult to separate, they were analyzed together. If paternal mitochondria-derived signals could not be separated from the MO-derived signals, they were excluded from analysis. After subtracting the background intensity derived from the cytoplasm of each zygote from the measured values, the fluorescence intensity of GFP was normalized to that of the paternal mitochondria or MOs in the same area because the position of the z-axis influenced the fluorescence intensity. The line profile of the localization of GFP-ALLO-1a and GFP-ALLO-1b was analyzed using Fiji software. To analyze the GFP-LGG-1 shape in Figs. 5 and 6, the autophagosome shapes were manually classified by referring to reference images.

## Immunostaining

For immunostaining, dissected zygotes or embryos were permeabilized using the freeze-crack method as previously described[6]. Briefly, a drop of M9 buffer containing 10–20 mM levamisole was placed on a glass slide. Into the droplet, 40–60 adult hermaphrodites at 24–36 h after the L4 stage were placed, dissected, covered with a cover glass,

and frozen at −80 °C for at least 10 h. Immediately before immunostaining, the cover glass of the frozen samples was vigorously removed to peel off fertilized eggshells. For immunostaining of ALLO-1a and ALLO-1b, zygotes were fixed in methanol for 5 min and then in acetone for 2 min at −20 °C. Fixed zygotes were dried at 25 °C to remove acetone, blocked with PTB (PBS containing 1 mM EDTA, 1% [w/v] bovine serum albumin, 0.05% [w/v] NaN₃, and 0.1% [v/v] Tween-20), and incubated with an anti-MO antibody 1CB4[35] (dilution 1:250) and anti-ALLO-1a (dilution 1:500) or anti-ALLO-1b antibodies (dilution 1:100) overnight at 4 °C. For observation of GFP-LGG-1 around MOs, zygotes were fixed in 3.2% paraformaldehyde solution (3.2% [w/v] paraformaldehyde, 0.24 M sorbitol, 400 mM PIPES, 20 mM EDTA, and 20 mM MgCl₂, pH 6.9) after immersion in methanol at −20 °C for 15 s. Fixed zygotes were blocked with PEMTB (0.1% [v/v] TritonX-100, 0.1% [w/v] bovine serum albumin, 400 mM PIPES, 20 mM EDTA, and 20 mM MgCl₂, pH 6.9) and incubated with the 1CB4 antibody (dilution 1:250) overnight at 4 °C. After washing with primary antibodies, the samples were incubated with the corresponding secondary antibodies, enclosed in SlowFade Diamond (Thermo Fisher Scientific) with 4′,6-diamidino-2-phenylindole (DAPI), and observed using FV1200. The following secondary antibodies were used for immunostaining: Goat anti-Rabbit IgG (H + L) Highly Cross-Adsorbed Secondary Antibody, Alexa Fluor Plus 488 (A-32731), Goat anti-Mouse IgG (H + L) Highly Cross-Adsorbed Secondary Antibody, Alexa Fluor 488 (A-11029), Goat anti-Rabbit IgG (H + L) Highly Cross-Adsorbed Secondary Antibody, Alexa Fluor 555 (A-21429), Goat anti-Mouse IgG (H + L) Highly Cross-Adsorbed Secondary Antibody, Alexa Fluor 555 (A-21424), Goat anti-Rat IgG (H + L) Cross-Adsorbed Secondary Antibody, Alexa Fluor 594 (A-11007), and Goat anti-Mouse IgG (H + L) Highly Cross-Adsorbed Secondary Antibody, Alexa Fluor Plus 647 (A-32728) (all antibodies are from Life Technologies, dilution 1:1000).

## Super-resolution microscopy

For observation by super-resolution microscopy, zygotes were fixed in 3.2% paraformaldehyde solution after immersion in methanol, as described above. Fixed zygotes were blocked with PEMTB and incubated with an anti-GFP (598, Medical and Biological Laboratories, dilution 1:400) and anti-mCherry antibody (M11217, clone 16D7, Thermo Fisher Scientific, dilution 1:100) overnight at 4 °C. After washing with primary antibodies, the samples were incubated with the corresponding secondary antibodies, enclosed in SlowFade Diamond (Thermo Fisher Scientific) with DAPI, and observed using an FV3000-Olympus Super Resolution system (FV3000-OSR, Olympus), which is based on the algorithm developed for spinning disk super-resolution microscopy[58]. We used 100× 1.45NA UPLXAPO100XO (Olympus) to obtain images. For the 3D projection and 3D reconstruction in Supplementary Movie 3, noise reduction was performed and then the images were projected to create 3D projection images using Fiji software. The 3D reconstruction was performed using 3D Viewer in Fiji software.

## Time-lapse imaging

We improved a method to immobilize adult hermaphrodites for live imaging. Adult hermaphrodites at 24−36 h after the L4 stage were incubated with levamisole (0.5−1.5 mM) in M9 buffer for 10 min on a 10% agarose pad surrounded by a 0.5 mm thick silicone rubber frame. After incubation, a cover glass was placed over the worms. Samples were observed using an IX73 (Olympus) equipped with a 100 × 1.35 NA UPlanSApo, 60 × 1.30 NA silicone objective lens, or a 60 × 1.40 NA UplanSApo oil-objective lens, an EMCCD camera iXon X3 (Andor Technology, Belfast, Northern Ireland), and a confocal scanner unit CSU22 (Yokogawa, Tokyo, Japan). Images were acquired every 15 or 20 s at 1−1.5 μm intervals in the Z direction, and the maximum intensity projection was created using Micro-Manager software[59] or Andor iQ (Oxford Instruments).

## TMT analysis

Fertilized eggs (wild type, *ikke-1*, or *allo-1* in four, three, or three biological replicates, respectively) were collected from approximately 10,000−20,000 adult hermaphrodites, equivalent to 4−7 plates of 10-cm dish culture, at 60−72 h after the L1 stage using the bleaching method, as previously described[56]. Briefly, the worms were collected using M9 buffer, sterilized by autoclaving, and mixed with an equal amount of bleaching solution (1 N NaOH, 0.5% [v/v] sodium hypochlorite). Fertilized eggs were collected by centrifugation after the nematode bodies were dissolved. Eggs were suspended in 6 M guanidine-HCl, 100 mM Tris-HCl (pH 8.0), and 2 mM dithiothreitol and immediately frozen in liquid nitrogen. The thawed lysates were dissolved through heating and sonication, followed by centrifugation at 20,000 × g for 15 min at 4 °C. The supernatants were reduced in 5 mM dithiothreitol at room temperature for 30 min and alkylated in 27.5 mM iodoacetamide at room temperature for 30 min in the dark. Proteins (400 μg each) were purified using methanol-chloroform precipitation and solubilized with 50 μL of 0.1% RapiGest SF (Waters Corporation, Milford, MA, USA) in 50 mM triethylammonium bicarbonate. The proteins were digested with 4 μg of Trypsin/Lys-C mix (Promega) for 16 h at 37 °C. Peptide concentrations were determined using a Pierce quantitative colorimetric peptide assay (Thermo Fisher Scientific). Approximately 150 μg of peptides for each sample was labeled with 0.2 mg of TMT-10plex reagents (Thermo Fisher Scientific) for 1 h at 25 °C. After the reaction was quenched with hydroxylamine, all TMT-labeled samples were pooled, acidified with trifluoroacetic acid (TFA), and analyzed using a High-Select Fe-NTA phosphopeptide enrichment kit (Thermo Fisher Scientific). The eluates were acidified and fractionated using a Pierce High pH reversed-phase peptide fractionation kit (Thermo Fisher Scientific) according to the manufacturer's instructions. Eight fractions were collected using 5%, 7.5%, 10%, 12.5%, 15%, 17.5%, 20%, and 50% acetonitrile (ACN). Each fraction was evaporated using a SpeedVac concentrator and dissolved in 0.1% TFA.

Liquid chromatography tandem mass spectrometry (LC-MS/MS) analysis of the resulting peptides was performed on an EASY-nLC 1200 UHPLC system connected to a Q Exactive Plus mass spectrometer using a nanoelectrospray ion source (Thermo Fisher Scientific). The peptides were separated on a C18 reversed-phase column (75 μm × 150 mm; Nikkyo Technos, Tokyo, Japan) with a linear gradient of 4−20% ACN for 0−150 min and 20−32% ACN for 150−190 min, followed by an increase to 80% ACN for 10 min and a final hold at 80% ACN for 10 min. The mass spectrometer was operated in data-dependent acquisition mode using the top 10 MS/MS method. The MS1 spectra were measured at a resolution of 70,000, an automatic gain control (AGC) target of 3e6, and a mass range of 375−1400 $m/z$. HCD MS/MS spectra were acquired at a resolution of 35,000, AGC target of 1e5, isolation window of 0.7 $m/z$, maximum injection time of 100 ms, and normalized collision energy of 32. The dynamic exclusion was set at 30 s. Raw data were directly analyzed against the *C. elegans* WormBase protein database using Proteome Discoverer 2.1 (Thermo Fisher Scientific) with the Sequest HT search engine for identification and TMT quantification. The search parameters were as follows: (a) trypsin as an enzyme with up to two missed cleavages, (b) precursor mass tolerance of 10 ppm, (c) fragment mass tolerance of 0.02 Da; (d) TMT of lysine and peptide N-terminus and carbamidomethylation of cysteine as fixed modifications, and (e) oxidation of methionine, deamidation of asparagine and glutamine, and phosphorylation of serine, threonine, and tyrosine as variable modifications. Peptides were filtered at a false-discovery rate of 1% using a percolator node.

## LC-MS/MS analysis to identify EPG-7 phosphorylation sites

**Sample preparation.** Fertilized eggs (wild type or *ikke-1* mutant) were collected from approximately 10,000 or 70,000 adult hermaphrodites, equivalent to 5 or 35 plates of 10-cm dish culture, at 60−72 h after

the L1 stage using the bleaching method[56] as described above, and immediately frozen in liquid nitrogen. The eggs were thawed on ice in HEPES-RIPA3 buffer (20 mM HEPES-KOH pH 7.4, 150 mM NaCl, 1% [v/v] NP40, 0.25% [w/v] Na-deoxycholate, 0.05% [w/v] SDS, and 50 mM NaF) with Complete EDTA-free Protease Inhibitor Cocktail (Roche, Basel, Switzerland) and PhosSTOP (Roche). The eggs were sonicated, incubated for 15 min at 4 °C, and centrifuged at 16,200 × g for 15 min at 4 °C. The supernatant was collected and centrifuged at 16,200 × g for 10 min at 4 °C. The supernatant was transferred to a fresh tube, and GFP-Trap magnetic agarose beads (Proteintech, Rosemont, IL, USA) that had been prewashed with HEPES-RIPA3 buffer were added. The beads and supernatant were rotated for 2 h at 4 °C. After rotation, the beads were transferred to a Protein LoBind tube (Eppendorf, Hamburg, Germany), washed three times with HEPES-RIPA3 buffer, and washed twice with 50 mM ammonium bicarbonate. The beads were divided to three new Protein LoBind tube with 50 μL of 50 mM ammonium bicarbonate. The proteins on the beads were digested using three methods. A) Beads and 200 ng of trypsin/LysC mix (Promega) were mixed and shaken overnight at 37 °C. B) Beads and 200 ng of chymotrypsin (Roche) were mixed and shaken with 10 mM CaCl₂ overnight at 25 °C. C) Beads and 40 ng of Asp-N (Promega) were mixed and shaken overnight at 37 °C. After the reaction, supernatants containing the digested peptides were collected, reduced, alkylated, acidified to pH 2.5 with TFA, and desalted using GL-Tip SDB (GL Sciences, Tokyo, Japan). The eluates were evaporated and dissolved in 3% ACN and 0.1% TFA.

**LC-MS/MS analysis.** LC-MS/MS analysis of the resulting peptides was performed on an EASY-nLC 1200 UHPLC system connected to an Orbitrap Fusion mass spectrometer using a nanoelectrospray ion source (Thermo Fisher Scientific). The peptides were separated on a C18 reversed-phase column with a linear 4–32% ACN gradient for 0–100 min, followed by an increase to 80% ACN for 10 min, which was held at 80% ACN for 10 min. The mass spectrometer was operated in data-dependent acquisition mode with a maximum duty cycle of 3 s. The MS1 spectra were measured at a resolution of 60,000, an AGC target of 4e5, and a mass range of 350–1500 m/z. HCD MS/MS spectra were acquired using an Orbitrap with a resolution of 30,000, AGC target of 5e4, isolation window of 1.6 m/z, maximum injection time of 54 ms, and normalized collision energy of 30. Dynamic exclusion was set to 15 s. Raw data were directly analyzed against the C. elegans WormBase protein database supplemented with the EPG-7-GFP sequence using Proteome Discoverer 2.4 (Thermo Fisher Scientific) with the Sequest HT search engine. The search parameters were as follows: (a) trypsin, chymotrypsin, or Asp-N as an enzyme with up to two, three, or three missed cleavages, respectively; (b) precursor mass tolerance of 10 ppm; (c) fragment mass tolerance of 0.02 Da; (d) carbamidomethylation of cysteine as a fixed modification; (e) acetylation of the protein N-terminus, oxidation of methionine, and phosphorylation of serine, threonine, and tyrosine as variable modifications. Peptides were filtered at a false discovery rate of 1% using percolator node. Label-free quantification was performed based on the intensities of the precursor ions using a precursor-ion quantifier node.

**Statistical analyses and reproducibility**
Individual plots were drawn using GraphPad Prism 8 software (GraphPad, Inc., La Jolla, CA, USA). Data were analyzed using a two-tailed unpaired t-test, Kruskal–Wallis with Steel-Dwass pairwise comparison test, one-way ANOVA with TukeyHSD pairwise comparison test, or two-tailed Mann–Whitney U test using GraphPad Prism 8 software or EasyR[60]. Data are provided as scatter plots. In the scatter plots, the bars indicate the mean ± standard deviation (SD). No statistical method was used to predetermine sample size. No data were excluded from the analyses. The experiments were not randomized, and the investigators were not blinded to allocation during experiments and outcome assessment. All strains were observed at least three times with similar results. All experiments were performed at least twice, and similar results were obtained.

**Reporting summary**
Further information on research design is available in the Nature Portfolio Reporting Summary linked to this article.

## Data availability
The MS proteomics data have been deposited to the ProteomeXchange consortium via the jPOST partner repository with the dataset identifiers PXD048126 (Fig. 8f, g) and PXD048129 (Supplementary Table 3). Source data for graphs (Figs. 2b, d, 4b, 5b, e, 6b, e, 8b, d, e) and blots (Fig. 7e) are provided with this paper as a Source Data file and Supplementary Fig. 7e, respectively. The imaging data supporting the findings of this study are available from the authors upon reasonable request. Source data are provided with this paper.

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

## Acknowledgements

We thank Kohei Nishino, Naomi Terawaki, Mei Tajima, Katsuya Sato, Yuhkoh Satouh, and the members of M. Sato and K. Sato laboratories for their technical assistance and discussion. We are grateful to Steven L'Hernault, Barth D Grant, Yuji Kohara, and Shohei Mitani for supplying the reagents and strains used in this study. Several strains were provided by the Caenorhabditis Genetic Center, which is funded by the NIH Office of Research Infrastructure Programs (P40 OD010440), and National Bioresource Project for the Experimental Animal "Nematode *C. elegans*". This study was supported by the Japan Society for the Promotion of Science KAKENHI grants (grant numbers 19H05712 and 21H02472 to M.S., 19H05711 and 20H00466 to K.S., 20J01777 and 22K15097 to T.S., and 22KJ0444 to T.N.), Takeda Science Foundation grant (to M.S.), Joint Usage and Joint Research Program of the Institute of Advanced Medical Sciences at Tokushima University, and the Institute for Molecular and Cellular Regulation at Gunma University.

## Author contributions

M.S., T.S., Y.K., and K.S. designed the experiments. T.S., Y.K., T.N., and M.S. performed the experiments. H.K. performed mass spectrometry analysis. M.S. and K.S. supervised this study. T.S., M.S., K.S., T.N., and H.K. wrote the manuscript.

## Competing interests

The authors declare no competing interests.
