## [Peer Review File · Nature Communications]

ALLO-1- and IKKE-1-dependent positive feedback mechanism promotes the initiation of paternal mitochondrial autophagyREVIEWER COMMENTS

Reviewer #1 (Remarks to the Author):

General comments: The authors found for the first time that ALLO-1, an upstream autophagy-related protein, regulates the recruitment of the autophagy membrane initiation complex and established that IKKE-1 plays a positive regulatory role in this process through kinase activity (phosphorylation) and is thus involved in the elimination of parental mitochondrial DNA during nematode fertilized egg development. It is well documented that paternal mitochondrial DNA is eliminated both before and after fertilization, whereas paternal mtDNA is degraded by selective autophagy in the egg after fertilization in nematodes. The results of this manuscript are complementary to the mechanism of elimination of paternal mtDNA in nematodes after fertilization. The major problem with the paper is that if paternal sperm elimination depends on autophagy, knocking down anyone of autophagy core genes would theoretically affect paternal sperm elimination. Because ALLO -1 is an upstream protein in the complex that regulates autophagy initiation, it will always have a greater or lesser impact on paternal mitochondrial DNA elimination, so the authors' findings lack novelty from this perspective. The data in the article are largely based on morphological descriptions (mainly fluorescence images), without sufficient alternative data to support the conclusions drawn by the authors.

Minor points:

1. The B-I representation in the figure legend of Fig. 1 does not match the image.
2. Why does ALLO -1b C-term in Fig. 6 A and D11A and E12A in Fig. 6 C show weak positivity? Please provide an explanation.
3. In Supplementary Movie 1, GFP-ALLO -1a shows co-localization with the parental mtDNA in the fertilization vesicle, but the text does not elaborate on this phenomenon.
4. In Supplementary Movie 1, GFP-ALLO -1a is also recruited to parental mtDNA in the fertilized oocyte, does this correlate with the recruitment of GFP-ALLO -1b? The authors need to validate the rescue experiments in IKKE-1 mutants as well as ALLO1 mutants that replenish both isoforms in order to perform the imaging experiments on autophagosome expansion in fertilized eggs.
5. New experiments need to be added to follow up on the autophagosome expansion imaging experiments in which tri-color fluorescence (both isoforms of ALLO1 plus the parental mtDNA) is observed in fertilized nematode eggs.
6. Evidence that ALLO -1b (but not ALLO -1a) is involved in the elimination of paternal mtDNA from fertilized nematode eggs is insufficient and needs to be elucidated by new and more definitive experiments. For example, by what mechanism do the differences in the C-terminal of the two ALLO -1 isoforms cause them to recognize different cargoes?
7. According to the authors, ALLO -1 is recruited to the parental mtDNA in the fertilized egg to mediate elimination. Biochemical experiments on the interaction of ALLO -1 with parental mtDNA are needed to address this question (direct interaction? Indirect interaction? Specific interaction sites?)
8. The authors hypothesize that IKKE-1 kinase activity impairs the ability of autophagy proteins such as ALLO -1 and EPG-7 to recruit around parental mtDNA. Please demonstrate experimentally that IKKE-1 kinase activity inhibits the recruitment of the proteins ALLO -1 and EPG-7 and not the transcription of the proteins, and that the reduction in the recruitment of the proteins is caused by the inhibition.

9. What do the authors make of the fact that loss of IKKE-1 kinase activity attenuates rather than completely inhibits the recruitment of autophagy-related proteins? Is there another synergistic mechanism that works (eg, the presence of different isoforms or other proteins that replace the kinase activity of IKKE-1)? Authors are invited to contribute to the discussion or add relevant experiments.

Reviewer #2 (Remarks to the Author):

The authors' group has been investigating the mechanism of paternal mitochondria elimination in *C. elegans* embryos and identified the allophagy adaptor ALLO-1 and its regulator IKKE-1/TBK1. The present paper represents a significant extension of their previous studies.

It is well-established that cargo can induce autophagy. One of the mechanisms is the binding between cargo adaptors and the ATG1/ULK complex. This binding is observed not only with soluble autophagy adapters such as NDP52 and OPTN but also with the ER-phagy receptor CCPG1 and yeast proteins such as Atg19, 30, 32, 36, and 39. This study by Sato's group clearly demonstrates that this concept applies to the mitophagy adaptor ALLO-1 in *C. elegans*, solidifying the mechanism of how cargo adaptors induce autophagy. The paper further presents the evidence for positive regulation of EPG7/ATG11 by IKKE-1/TBK1 and the existence of a positive feedback loop through the IKK-1–EPG-7 axis.

The paper is timely and well-written, with clear data. This reviewer has only a few minor comments.

1. Figure 1 shows that ALLO-1a primarily localizes to MOs, while ALLO-1b localizes predominantly to mitochondria. However, in Figure 2, the mitochondrial localization of ALLO-1a seems more pronounced (at least initially). Additionally, ALLO-1a appears to be significantly involved in paternal mitochondrial removal (Figure 1K). To make these data more understandable, the authors should consider including some form of quantification for the colocalization of the two ALLO-1 isoforms with paternal mitochondria and MOs.

2. Regarding the phospho-proteome analysis in Figure 7, it would be helpful to provide a comment on whether ALLO-1 was identified as a protein whose phosphorylation changes depending on IKKE-1.

3. Reference #29 shows that OPTN recruits TBK1, and NDP52 recruits FIP200. ALLO-1 plays a dual role in recruiting both EPG7/ATG11 and IKKE-1/TBK1. Could the authors add some comments on whether ALLO-1 has both OPTN- and NDP52-like functions?

Reviewer #3 (Remarks to the Author):

In this manuscript, Sasaki et al. examined autophagosome formation during allophagy which is the selective removal of paternally inherited mitochondria and other sperm-derived organelles. In particular,

the authors sought to mechanistically dissect the roles of the allophagy adaptor ALLO-1 and the kinase IKKE-1 using *C. elegans* as model. Briefly, the authors unveiled that ALLO-1 exists in two different isoforms, ALLO-1a and ALLO-1b which differ in their C-terminus and their cargo specificity. Using time-lapse imaging, the authors found that both ALLO-1 isoforms localize to allophagy cargo well before other autophagy components including the autophagosome marker LGG-1 and that the recruitment of the kinases UNC-51 and IKKE-1 depends on ALLO-1. Intriguingly, the authors uncovered a feedback loop in which ALLO-1 is rapidly recruited to cargo independent of IKKE-1 but further accumulation of ALLO-1 is blocked when IKKE-1 was absent or its kinase activity inhibited. Further, the authors provide evidence that the kinase function of IKKE-1 contributes to the expansion of autophagosomal membranes around different types of allophagy cargo. Next, the authors showed that ALLO-1 binds the UNC-51 adaptor EPG7 (a homolog of FIP200) in a manner dependent on its well-conserved FIP200-interacting region (FIR) and that this interaction is required to drive allophagy. Interestingly, the authors uncovered that EPG7 is part of another ALLO-1 accumulation regulating feedback mechanism, possibly controlled by IKKE-1 phosphorylation. Overall, the work of Sasaki and colleagues represents an elegant and comprehensive study which provides important mechanistic insights into allophagy in worms. Therefore, only a few minor points need to be addressed.

- 1) Regarding the phospho-proteomics data in Figure 7: It would be insightful if the authors add a scheme of EPG-7 (and other interesting allophagy components) with the identified phospho-sites. If an EPG-7 structure is available, the authors should use this to model the phospho-sites.
- 2) Is the C-terminal divergent region of ALLO-1a and ALLO-1b sufficient to localize to paternally inherited mitochondria and MOs, respectively?
- 3) Does swapping of the divergent C-terminus of ALLO-1a and ALLO-1b changes the localization pattern?
- 4) Given that IKKE-1 controls ALLO-1 accumulation, it is somewhat surprising that the phospho-proteomics did not identify phospho-sites in ALLO-1.

Reviewer #4 (Remarks to the Author):

The authors investigated how ALLO-1, IKKE-1 and EPG-7 function together to mediate allophagy in *C. elegans* embryos. They report that ALLO-1 possesses two isoforms that differ in their C-terminal part, which exhibit different distributions and functions, with ALLO-1a localizing mainly to sperm-derived Membranous Organelles (MOs), ALLO-1b to sperm-derived mitochondria. Live imaging experiments established that GFP-ALLO-1b is recruited to paternal mitochondria 30 s after sperm-oocyte contact, whereas GFP-ALLO-1a is recruited 2 min 30 s later on average, with GFP-LGG-1 recruitment following only minutes later. Next, the authors investigated whether ALLO-1 is needed for recruiting the ULK1 complex components UNC-51 and EPG-7, and found this to be the case. Moreover, IKKE-1 distribution around cargo is severely impaired in *allo-1* null mutants. Interestingly in addition, IKKE-1, as well as its

kinase activity, are necessary for the full presence of ALLO-1b at paternal mitochondria. The authors also tested the requirements of IKEE-1, finding that it is needed for expanding the autophagosome membrane as judged by GFP-LGG-1 signal intensity, with a more pronounced requirement for paternal mitochondria than for MOs. The authors then report yeast two hybrid experiments demonstrating that ALLO-1 associates with EPG-7 through a so-called FIR motif. Mutations in this motif abolish interactions in yeast two-hybrid, in a cell-free association assay, as well as in the worm. Furthermore, the authors find that ALLO-1 accumulation is partially defective in an *epg-7* mutant allele. In a last set of experiments, the authors sought to identify how IKEE-1-mediated phosphorylation orchestrates allopagy by conducting a comprehensive proteomic experiment, identifying notably a peptide of EPG-7 containing S949 as a promising lead. However, mutating this residue to alanine did not abolish paternal mitochondrial degradation.

This is a potentially interesting study that advances understanding of the mechanisms of allopagy in *C. elegans*. Although many of the conclusions are well supported by the data, some important points need to be clarified further before publication can be envisaged, as explained below.

Most important points

- One general issue pertains to when exactly during development some of the analyses are conducted for each embryo. This is critical given that some of the effects are rather small, and that the signal intensity of GFP-ALLO-1 increases during the cell cycle. For example, in Fig. 5A-B, the authors need to make sure that the small diminution in GFP-ALLO-1 levels in the *ikke-1* null is not due simply to scoring embryos as a slightly earlier stage (related to this: please adjust brightest and contrast for the DAPI signal or else report it separately, as the current rendition does not allow one to assess developmental timing). A similar comment holds for Fig. S4A-B.

- The authors should quantify EPG-7-GFP distribution in the *allo-1* null allele (Fig. S3). The author write on p. 6 that the “localization... was almost completely inhibited”. This is a central point because the postulated feedback mechanism schematized in Fig. 7H and featured in the title of the paper requires that NO EPG-7 or IKEE-1 be present in an *allo-1* null mutant. Otherwise parallel pathways would offer a more plausible explanation for how the system operates.

- The impact of IKKE-1 on ALLO-1 distribution is an important cornerstone of this study, and Fig. 3A-B clearly shows that IKKE-1 is critical for GFP-ALLO-1b recruitment to paternal mitochondria. However, no quantification is provided in Fig. S4 to make sure that the same holds for endogenous ALLO-1b. Reading the legend of the figure -which refers to a panel D that is not shown- perhaps the quantification has been conducted but somehow forgotten in the submitted version of the manuscript?

- How can the authors ascertain whether GFP-LGG-1 surrounds paternal mitochondria partially or entirely without super-resolution microscopy and 3D reconstruction? Also, the authors should not only show schematics in Fig. 4D but also an exemplary image for each category so that readers can have a better sense of what the classification is based on.

Other points

- The authors performed an impressive phosphoproteomic analysis to try to identify the relevant

residue(s) phosphorylated by IKKE-1 during allophagy. Disappointingly, the S949A mutant did not seem to prevent paternal mitochondrial degradation (note, however, that there are no numbers nor quantifications in Fig. S6 to assess how firm this conclusion is). The authors should consider testing whether an S949D/E mutant can bypass the need for IKKE-1 as another way to address the potential relevance of S949 phosphorylation. Moreover, mutating all 11 phosphorylation sites could be envisaged especially since the experiments are conducted with transgenic strains.

- There appears to be a disconnect between Fig. 2B and Fig. 1. Indeed, Fig. 2B seems to indicate that most of the GFP-ALLO-1a signal (e.g. at 180'') is at paternal mitochondria, not MOs (which I assume are what the arrowheads point to, this is something the legend does not spell out), whereas this is not what Fig. 1 conveys.

- The authors generated transgenic lines by bombardment for their in vivo experiments. A potential caveat with such lines is that expression levels can vary between strains. Insertions by MosCI would seem like a better, more controlled, strategy.

- To further ascertain the specificity of the ALLO-1 antibodies towards their respective isoforms, the authors could conduct Western blot analysis as the two isoforms have a different size. Moreover, they could also conduct immunofluorescence experiments with the transgenic lines expressing solely ALLO-1a or solely ALLO-1b in the background of the allo-1 null.

- In Fig. 2, it was not clear why the authors did not conduct such experiments also with an MO marker (bearing mCherry for instance), to ascertain that ALLO-1a truly associates with MOs.

- Fig. S4B is lacking statistical analysis. The text on p. 7 mentions that "further accumulation was impaired", but Fig. S4B seems to indicate little change.

- The authors should indicate in the Materials and Methods what amounts of starting material (i.e. animals) were utilized for the proteomic analyses.

- Table S1 does not report phosphopeptides, as the title indicates, but merely a list of proteins in which phosphopeptides were found. This needs to be fixed and the actual sequences provided.

- The authors may want to comment on whether having two isoforms of ALLO-1 is a conserved feature amongst nematodes.

- Although the precise role of IKKE-1 was unclear before this work, is this also the case of its mammalian counterparts? This should be mentioned explicitly on p. 6.

Small points

- On p. 5, the authors mention that they "developed" a method for time-lapse imaging of allophagy, but reading the corresponding Materials and Methods section did not make it clear what aspects -beyond standard live imaging approaches- were developed specifically for this purpose.

- Fig. S2: note that pronuclei do not fuse in *C. elegans* when they meet, they merely appose. Also, therefore, the schematic in mitosis should represent two pronuclei, not just one, and the text also be rectified in a couple of places accordingly (on p. 6 for instance). Also, at what temperature is the time line? This should be indicated.

- On p. 7, the authors write that "... GFP-LGG-1 puncta were occasionally attached to 23-32% of paternal mitochondria...". This is confusing: grammatically, this seems to indicate that occasional attachment was observed in these 23-32%, but what about the others? Or do the authors mean to convey that attachment was observed in this percentage of embryos? Please clarify.

- When the authors write on p. 10 that "ALLO-1 is the most upstream regulator", they should write instead "ALLO-1 is the most upstream regulator known", as other component yet to be discovered might

take on that role in the future.

- What is an FV1200 microscope system (p. 18)? The authors should provide more explanation.
- On p. 19, the authors list three lenses with which time-laps imaging was conducted apparently. If so, the term "or" seems to be missing between the mention of the lenses.
- Throughout the figures: by convention, embryos should be shown with the anterior to the left.
- Fig. 7F-G: the legend is missing on the X axis.
- Fig. S5, typo: H. sapiens (no capital S).

Response to reviewers' comments

Comment from Reviewer #1

General comments: The authors found for the first time that ALLO-1, an upstream autophagy-related protein, regulates the recruitment of the autophagy membrane initiation complex and established that IKKE-1 plays a positive regulatory role in this process through kinase activity (phosphorylation) and is thus involved in the elimination of parental mitochondrial DNA during nematode fertilized egg development. It is well documented that paternal mitochondrial DNA is eliminated both before and after fertilization, whereas paternal mtDNA is degraded by selective autophagy in the egg after fertilization in nematodes. The results of this manuscript are complementary to the mechanism of elimination of paternal mtDNA in nematodes after fertilization. The major problem with the paper is that if paternal sperm elimination depends on autophagy, knocking down anyone of autophagy core genes would theoretically affects paternal sperm elimination. Because ALLO -1 is an upstream protein in the complex that regulates autophagy initiation, it will always have a greater or lesser impact on paternal mitochondrial DNA elimination, so the authors' findings lack novelty from this perspective. The data in the article are largely based on morphological descriptions (mainly fluorescence images), without sufficient alternative data to support the conclusions drawn by the authors.

Thank you for the time and effort expended on reviewing our manuscript. The reviewer suggested that our study largely recapitulates published findings. We apologize for not clearly stating the novelty of this study. However, we believe that it does not reflect the multiple novel findings reported in our study. We previously reported that paternal mitochondria and their mitochondrial (mt)DNA are eliminated from embryos through autophagy, and identified ALLO-1 and IKKE-1 as key regulators of autophagy. However, how these factors regulate the local and selective formation of autophagosomes around sperm-derived organelles, including paternal mitochondria and membranous organelles (MOs), remains largely unknown. In the present study, using our live imaging system, we first revealed the extremely rapid localization of ALLO-1 and IKKE-1 to paternal organelles within minutes after fertilization, following the recruitment of LGG-1 (LC3 homolog) approximately 5 min after fertilization. We also determined that ALLO-1 directly bound to EPG-7 and regulated the recruitment of the ATG1/ULK1 initiation complex to the target. In addition, we showed a positive feedback loop between ALLO-1

and EPG-7. To date, it is unclear how IKKE-1, a TBK1-related kinase, is involved in autophagy. In this study, we elucidated that the kinase activity of IKKE-1 further drives this positive feedback, and the accumulation of ALLO-1 and EPG-7 around the cargo is required for the initiation of autophagosomal membrane formation. Several mammalian autophagy adaptors have been reported to bind to FIP200, a homolog of EPG-7, and directly recruit the ULK1 complex in cultured cells (Vargas, et al., 2019; Turco, et al., 2019; Ravenhill, et al., 2019). It is also known that TBK1 is involved in selective autophagy (Vargas, et al., 2022). However, the precise relationship between each factor remains to be fully understood, especially under physiological conditions such as in animal models. We believe that our findings show novel aspects of developmentally regulated selective autophagy that takes place under physiological conditions, but also advances our understanding of how selective autophagy is initiated around cargo and how the autophagy adaptor and TBK1-related kinase are involved. Our results also suggest that the fundamental mechanisms of selective autophagy are well-conserved in other species.

We thank the reviewers for their comments and suggestions. This has assisted us in improving the manuscript. Please refer to our point-by-point responses below:

Minor points:

1. The B-I representation in the figure legend of Fig. 1 does not match the image.

Response to 1: We thank the reviewer for this comment. We have corrected the alphabet in Figure 1.

2. Why does ALLO -1b C-term in Fig. 6 A and D11A and E12A in Fig. 6 C show weak positivity? Please provide an explanation.

Response to 2: We agree that some colonies expressing ALLO-1b C-terminus and D11A or E12A single mutants showed faint LacZ expression and these ALLO-1 mutants may still weakly bind to EPG-7. However, the expression level of LacZ was clearly reduced in these mutants compared to that in wild-type ALLO-1. In addition, the D11A E12A double mutation strongly suppressed the interaction with EPG-7, and this result was confirmed through an in vitro binding assay. Therefore, we concluded that both D11 and E12 are important for EPG-7 binding.

3. In Supplementary Movie 1, GFP-ALLO -1a shows co-localization with the parental

mtDNA in the fertilization vesicle, but the text does not elaborate on this phenomenon.

Response to 3: We apologize that our explanation was not sufficiently clear and caused some confusion. First, we confirm that we monitored whole paternal mitochondria using an mCherry-tagged mitochondrial matrix protein but not paternal mtDNA in these experiments. Because ALLO-1 is a cytosolic protein without a mitochondrial targeting sequence or transmembrane domain, we assume that ALLO-1 proteins are peripherally targeted to the surface of paternal organelles after fertilization but are not transported inside these organelles. We have revised the Introduction section to explain ALLO-1 and organelle markers in detail. Second, we originally described this point on p. 6, line 20, as follows: “Surprisingly, time-lapse imaging revealed that GFP-ALLO-1a and GFP-ALLO-1b began to localize around the paternal mitochondria within 30 s of sperm–oocyte contact”. However, since ALLO-1a shows dynamic behavior and changes its localization during development, it may also cause some confusion. As originally mentioned on p. 6 line 20 and line 25, we observed that ALLO-1a first appeared on paternal mitochondria within approximately 30 s after fertilization, although the signal was weak. In addition to this paternal mitochondrial localization, ALLO-1a was observed on structures outside the paternal mitochondria, presumably MOs, approximately 3 min after sperm–oocyte contact (Fig. 3b (original Fig. 2b) and Supplementary Movie 1). The fluorescence intensity of these extramitochondrial ALLO-1a puncta increased further and became brighter than that of the paternal mitochondria. In Fig. 1, zygotes around the pronuclear expansion to the pseudo-cleavage stage (25–40 min after fertilization) were observed; therefore, ALLO-1a predominantly localized on MOs, but was also located around paternal mitochondria to some extent. This observation is consistent with the results of live imaging. We revised our statement to explain these results more carefully. We have also included new Supplementary Fig.1 to show that ALLO-1a weakly localizes to paternal mitochondria, in addition to MOs.

4. In Supplementary Movie 1, GFP-ALLO -1a is also recruited to parental mtDNA in the fertilized oocyte, does this correlate with the recruitment of GFP-ALLO -1b? The authors need to validate the rescue experiments in IKKE-1 mutants as well as ALLO1 mutants that replenish both isoforms in order to perform the imaging experiments on autophagosome expansion in fertilized eggs.

Response to 4: This was an important comment. As shown in Supplementary Movie 1, GFP-ALLO-1a was expressed in the *allo-1* deletion mutant lacking both ALLO-1 isoforms. Therefore, we believe that ALLO-1a weakly localizes to the paternal mitochondria independently of ALLO-1b.

In addition, the *ikke-1* mutant is defective in the degradation of paternal organelles, although endogenous ALLO-1a and ALLO-1b are both expressed, suggesting that the *ikke-1* mutant is not rescued by the simultaneous expression of ALLO-1a and b (Sato et al., 2018). We also found that expression of GFP-ALLO-1, which expressed both isoforms under germ line-specific *pie-1* promoter control, did not rescue the allophagy defects of the *ikke-1* deletion mutant as shown below.

5. New experiments need to be added to follow up on the autophagosome expansion imaging experiments in which tri-color fluorescence (both isoforms of ALLO1 plus the parental mtDNA) is observed in fertilized nematode eggs.

Response to 5: We thank the reviewer for the suggestion. In accordance with this comment, we attempted to detect paternal mtDNA by staining with fluorescent dyes such as SYBR Green, SYTOX Orange, or SiR-Hoechst. Unfortunately, dyes for live imaging of paternal mtDNA were unavailable. The amount of paternal mtDNA may be significantly small, or these dyes may not be able to permeate sperm or fertilized eggs. Instead, we compared the subcellular localization of ALLO-1a, ALLO1b, and paternal mitochondria using tricolor imaging of fixed zygotes. For this experiment, the wild-type zygotes expressing GFP-ALLO-1a were stained with an anti-ALLO-1b antibody. We confirmed that GFP-ALLO-1a and ALLO-1b partially overlapped around paternal mitochondria (arrow heads), although we also observed ALLO-1a-positive and ALLO-1b negative puncta (arrows), presumably MOs as shown below.

◀ Paternal mitochondria with anti-ALLO-1b signal
(+ slight GFP-ALLO-1a signal)

\ Estimated MO with GFP-ALLO-1a signal, but no anti-ALLO-1b signal

6. Evidence that ALLO-1b (but not ALLO-1a) is involved in the elimination of paternal mtDNA from fertilized nematode eggs is insufficient and needs to be elucidated by new and more definitive experiments. For example, by what mechanism do the differences in the C-terminal of the two ALLO -1 isoforms cause them to recognize different cargoes?

Response to 6: We agree that it is important to note how the C-terminal regions of ALLO-1 isoforms cause their different behaviors. To examine whether the C-terminal divergent regions of ALLO-1 are sufficient for the localization, we generated transgenic animals expressing GFP fused to the C-terminal region of ALLO-1a (355-388) or b (355-402). We found that both fusions were not detected on the paternal mitochondria and MOs. Because the C-terminal half of ALLO-1b (181-402) was localized to the paternal mitochondria, these results suggest that the C-terminal common region (181–354) is also involved in localization or proper protein folding. These new results have been included in the revised manuscript (revised Supplementary Fig. 3). Thus, we need further study to address the mechanism of ALLO-1 localization and are now investigating this point in a future study. In the present study, we focused on the molecular mechanisms by which ALLO-1, IKKE-1, and core autophagy regulators regulate autophagosome formation.

7. According to the authors, ALLO -1 is recruited to the parental mtDNA in the fertilized

egg to mediate elimination. Biochemical experiments on the interaction of ALLO -1 with parental mtDNA are needed to address this question (direct interaction? Indirect interaction? Specific interaction sites?)

Response to 7: As described in Response 3, as ALLO-1 is a cytosolic protein without a mitochondria-targeting signal or TMD, ALLO-1 proteins peripherally localize to the surface of paternal mitochondria and does not enter the interior of the mitochondria. Therefore, we assumed that ALLO-1 proteins did not directly bind to mtDNA for degradation. Instead, whole paternal mitochondria and their mtDNA are degraded through ALLO-1-mediated allopagy after fertilization. As mentioned in Response 6, we are attempting to elucidate how ALLO-1b localizes to paternal mitochondria in the next project. As the reviewer suggested, we are conducting some biochemical experiments, such as co-IP or proximity labeling, to identify the binding partner of ALLO-1, which would reveal a direct interaction of ALLO-1 and factors on the paternal mitochondria.

8. The authors hypothesized that IKKE-1 kinase activity impairs the ability of autophagy proteins such as ALLO -1 and EPG-7 to recruit around parental mtDNA. Please demonstrate experimentally that IKKE-1 kinase activity inhibits the recruitment of the proteins ALLO -1 and EPG-7 and not the transcription of the proteins, and that the reduction in the recruitment of the proteins is caused by the inhibition.

Response to 8: We hypothesized that IKKE-1 kinase activity enhances (does not impair) the recruitment of ALLO-1 and EPG-7 to the paternal mitochondria. In accordance with the this comment, we confirmed that the protein levels of GFP-tagged ALLO-1a and ALLO-1b were similarly expressed in the wild-type and *ikke-1* mutant backgrounds using immunoblotting experiments (revised Supplementary Fig.6a). We also quantified the fluorescence intensity of cytoplasmic EPG-7-GFP and confirmed that it did not change in *ikke-1* mutants (revised Supplementary Fig.6g). These results suggest that the weak recruitment of ALLO-1 and EPG-7 in the *ikke-1* mutant was not due to a reduction in protein levels.

9. What do the authors make of the fact that loss of IKKE-1 kinase activity attenuates rather than completely inhibits the recruitment of autophagy-related proteins? Is there another synergistic mechanism that works (eg, the presence of different isoforms or other proteins that replace the kinase activity of IKKE-1)? Authors are invited to

contribute to the discussion or add relevant experiments.

Response to 9: We apologize for the insufficient explanation. The loss of IKKE-1 results in partial, but not complete, inhibition of recruitment of autophagy factors, which was one of the primary findings of this study. Thus, we hypothesized the following: First, a small amount of ALLO-1 localizes to the cargo and then starts to recruit EPG-7 and IKKE-1. This initial step (referred to as the first step or cargo recognition step) was independent of IKKE-1. As ALLO-1 interacts with EPG-7 independently of IKKE-1 in our in vitro assay (revised Fig. 7e), we assumed that this direct interaction initiates the recruitment of EPG-7 to cargo even without IKKE-1. Thus, ALLO-1 and EPG-7 weakly localized to the cargo, even in the *ikke-1* mutant. However, the weak recruitment of ALLO-1 and EPG-7 is insufficient, and more molecules must be recruited to drive autophagosome formation (referred to as the second or accumulation step). We demonstrated that the kinase activity of IKKE-1 was involved in this second step. IKKE-1 may enhance and stabilize the interaction between ALLO-1 and EPG-7, as well as other autophagy factors, resulting in a positive feedback loop, such as the recruitment of EPG-7, resulting in the further accumulation of ALLO-1 molecules. Thus, the *ikke-1* mutant showed partial ALLO-1 and EPG-7 localization, whereas the *allo-1* mutant showed a complete loss of EPG-7 localization. Based on these results, we proposed a model to explain how IKKE-1 regulates selective autophagy. We have described our model clearly in the revised manuscript. However, we do not exclude the possibility that other kinases, such as UNC-51/ULK1, which are involved in allophagy, are also involved additively or sequentially to drive allophagy, as suggested by the reviewer. In mammals, it was recently reported that TBK1 and ULK1/2 have redundant functions in NDP52-mediated mitophagy (Nguyen et al., 2023). We have included this discussion in the revised manuscript as follows in p. 11:

“A recent study reported that TBK1 and ULK1 function redundantly in mammalian NDP52-mediated mitophagy, and mitophagy proceeds with only one of these kinases.³⁰ In contrast, both IKKE-1 and UNC-51 are required for normal allophagy.⁶
³¹ As allophagy is a rapid reaction compared to PINK1–Parkin-dependent mitophagy, the activation of both kinases may be necessary to drive such a reaction. IKKE-1 and UNC-51 may function additively or sequentially during allophagy.”

Comment from Reviewer #2

The authors' group has been investigating the mechanism of paternal mitochondria elimination in *C. elegans* embryos and identified the allophagy adaptor ALLO-1 and its regulator IKKE-1/TBK1. The present paper represents a significant extension of their previous studies.

It is well-established that cargo can induce autophagy. One of the mechanisms is the binding between cargo adaptors and the ATG1/ULK complex. This binding is observed not only with soluble autophagy adaptors such as NDP52 and OPTN but also with the ER-phagy receptor CCPG1 and yeast proteins such as Atg19, 30, 32, 36, and 39. This study by Sato's group clearly demonstrates that this concept applies to the mitophagy adaptor ALLO-1 in *C. elegans*, solidifying the mechanism of how cargo adaptors induce autophagy. The paper further presents the evidence for positive regulation of EPG7/ATG11 by IKKE-1/TBK1 and the existence of a positive feedback loop through the IKK-1–EPG-7 axis.

The paper is timely and well-written, with clear data. This reviewer has only a few minor comments.

1. Figure 1 shows that ALLO-1a primarily localizes to MOs, while ALLO-1b localizes predominantly to mitochondria. However, in Figure 2, the mitochondrial localization of ALLO-1a seems more pronounced (at least initially). Additionally, ALLO-1a appears to be significantly involved in paternal mitochondrial removal (Figure 1K). To make these data more understandable, the authors should consider including some form of quantification for the colocalization of the two ALLO-1 isoforms with paternal mitochondria and MOs.

Response to 1: We apologize for the lack of clarity in this explanation. As the comment pointed out, ALLO-1a localized to the paternal mitochondria in addition to MOs and rescued the defect in paternal mitochondrial degradation, although the efficiency was lower than that of ALLO-1b. Accordingly, we quantified the fluorescence intensity of ALLO-1a on paternal organelles and confirmed that ALLO-1a was detected on both paternal organelles, but more ALLO-1a was detected on MOs (revised Supplementary Fig. 1b). We have attempted to explain these results in detail in the revised manuscript. In addition, Figs. 1c and 3b (original Figs. 1b and 2b) may provide different impressions and cause misleading results. To detect the initial weak ALLO-1 signal in the paternal organelles, we used a relatively high laser power for

live imaging (Fig. 3). However, in zygotes at the later stage as shown in Fig. 1c, only a weak ALLO-1a signal was observed in the paternal mitochondria because the laser power was adjusted to the bright signal on the MOs. To avoid misunderstanding, we included additional images in which we compared the levels of ALLO-1a in MOs and paternal mitochondria (revised Supplementary Fig.1).

2. Regarding the phospho-proteome analysis in Figure 7, it would be helpful to provide a comment on whether ALLO-1 was identified as a protein whose phosphorylation changes depending on IKKE-1.

Response to 2: We thank the reviewer for this comment. Several phosphopeptides of ALLO-1 were identified using phosphoproteomic analysis, and their phosphorylation levels were reduced in the *ikke-1* mutant, although the change was not as drastic as that in EPG-7 S949. We have included this information in the revised Figure 8 and Supplementary Table 1.

3. Reference #29 shows that OPTN recruits TBK1, and NDP52 recruits FIP200. ALLO-1 plays a dual role in recruiting both EPG7/ATG11 and IKKE-1/TBK1. Could the authors add some comments on whether ALLO-1 has both OPTN- and NDP52-like functions?

Response to 3: We thank the reviewer for the suggestion. We have included this in the Discussion section of the revised manuscript as follows:

“In mammals, OPTN and NDP52 function redundantly in PINK1-Parkin-dependent mitophagy and directly bind to TBK1 and FIP200, respectively.^{21, 23, 42} Although OPTN and NDP52 are not conserved in *C. elegans*, ALLO-1 may have both OPTN- and NDP52-like functions.”

Comment from Reviewer #3

In this manuscript, Sasaki et al. examined autophagosome formation during allophagy which is the selective removal of paternally inhibited mitochondria and other sperm-derived organelles. In particular, the authors sought to mechanistically dissect the roles of the allophagy adaptor ALLO-1 and the kinase IKKE-1 using *C. elegans* as model. Briefly, the authors unveiled that ALLO-1 exists in two different isoforms, ALLO-1a and

ALLO-1b which differ in their C-terminus and their cargo specificity. Using time-lapse imaging, the authors found that both ALLO-1 isoforms localize to allophagy cargo well before other autophagy components including the autophagosome marker LGG-1 and that the recruitment of the kinases UNC-51 and IKKE-1 depends on ALLO-1. Intriguingly, the authors uncovered a feedback loop in which ALLO-1 is rapidly recruited to cargo independent of IKKE-1 but further accumulation of ALLO-1 is blocked when IKKE-1 was absent or its kinase activity inhibited. Further, the authors provide evidence that the kinase function of IKKE-1 contributes to the expansion of autophagosomal membranes around different types of allophagy cargo. Next, the authors showed that ALLO-1 binds the UNC-51 adaptor EPG7 (a homolog of FIP200) in a manner dependent on its well-conserved FIP200-interacting region (FIR) and that this interaction is required to drive allophagy. Interestingly, the authors uncovered that EPG7 is part of another ALLO-1 accumulation regulating feedback mechanism, possibly controlled by IKKE-1 phosphorylation. Overall, the work of Sasaki and colleagues represents an elegant and comprehensive study which provides important mechanistic insights into allophagy in worms. Therefore, only a few minor points need to be addressed.

1. Regarding the phospho-proteomics data in Figure 7: It would be insightful if the authors add a scheme of EPG-7 (and other interesting allophagy components) with the identified phospho-sites. If an EPG-7 structure is available, the authors should use this to model the phospho-sites.

Response to 1: We thank the reviewer for the suggestion. Based on the reviewer's suggestion, we have improved the scheme for the EPG-7 and FIP200 domain structures (revised Supplementary Fig. 8a). As EPG-7 has been reported to bind to several core autophagy regulators in different regions (Lin et al., 2013), this information was also included. We found at least 13 phosphorylation sites of EPG-7 in addition to S949. Notably, EPG-7 phosphorylation sites have been mapped to the oligomerization domain and binding region of other autophagy regulators. In addition, we attempted 3D structure prediction of EPG-7 and FIP200 by AlphaFold 2. Unfortunately, the predicted structures of FIP200 did not reflect the actual structure, because they are quite different from those observed using negative-stain electron microscopy (Shi, et al., 2020). Thus, we could not use AlphaFold2 for 3D structure prediction of EPG-7.

2. Is the C-terminal divergent region of ALLO-1a and ALLO-1b sufficient to localize to

paternally inherited mitochondria and MOs, respectively?

Response to 2: We thank the reviewer for the comment. As suggested by the reviewer, we examined whether the C-terminal region was sufficient for localization. We generated transgenic animals expressing GFP fused to the C-terminal divergent region of ALLO-1a (355-388) or b (355-402) and examined their localization. We determined that both fusions were not detected on the paternal mitochondria and MOs. Because the C-terminal half of ALLO-1b (181-402) was localized to the paternal mitochondria, these results suggest that the C-terminal common region (181–354) is also involved in localization or proper protein folding. These new results have been included in the revised manuscript (revised Supplementary Fig. 3).

3. Does swapping of the divergent C-terminus of ALLO-1a and ALLO-1b changes the localization pattern?

Response to 3: Because ALLO-1a and ALLO-1b share residues 1–354, swapping of the C-terminal region would render the same result. Instead, we examined whether the C-terminal divergent regions were sufficient for specific localization, as described above (Response 2). These results suggest that the C-terminal divergent regions are necessary for their different cargo preferences, but are insufficient. Because determining which part of ALLO-1 is sufficient for localization is an important question, we will continue this direction of study in a future study.

4. Given that IKKE-1 controls ALLO-1 accumulation, it is somewhat surprising that the phospho-proteomics did not identify phospho-sites in ALLO-1.

Response to 4: We thank the reviewer for the comment. The phosphorylation site of ALLO-1 was also identified and is indicated in revised Figure 8 and Supplementary Table 1. Several phospho-peptides of ALLO-1 were identified and their phosphorylation levels were reduced in the *ikke-1* mutant, although the change was not as drastic as that of EPG-7 S949. We have included this information in the revised Figure 8. Although the kinase activity of IKKE-1 is important for allopagy, we did not find any phosphorylation sites in EPG-7 or ALLO-1, with single mutations that have a drastic effect on allopagy. We also found that many downstream autophagy factors are phosphoproteins, and that the phosphorylation levels of some proteins were affected by the *ikke-1* mutation. As discussed in the main text p. 12, line 8~, we think

that the phosphorylation of multiple downstream proteins may synergistically enhance the accumulation of ALLO-1 and EPG-7, as well as downstream autophagy factors around the cargo.

Comment from Reviewer #4

The authors investigated how ALLO-1, IKKE-1 and EPG-7 function together to mediate autophagy in *C. elegans* embryos. They report that ALLO-1 possesses two isoforms that differ in their C-terminal part, which exhibit different distributions and functions, with ALLO-1a localizing mainly to sperm-derived Membranous Organelles (MOs), ALLO-1b to sperm-derived mitochondria. Live imaging experiments established that GFP-ALLO-1b is recruited to paternal mitochondria 30 s after sperm-oocyte contact, whereas GFP-ALLO-1a is recruited 2 min 30 s later on average, with GFP-LGG-1 recruitment following only minutes later. Next, the authors investigated whether ALLO-1 is needed for recruiting the ULK1 complex components UNC-51 and EPG-7, and found this to be the case. Moreover, IKKE-1 distribution around cargo is severely impaired in *allo-1* null mutants. Interestingly in addition, IKKE-1, as well as its kinase activity, are necessary for the full presence of ALLO-1b at paternal mitochondria. The authors also tested the requirements of IKKE-1, finding that it is needed for expanding the autophagosome membrane as judged by GFP-LGG-1 signal intensity, with a more pronounced requirement for paternal mitochondria than for MOs. The authors then report yeast two-hybrid experiments demonstrating that ALLO-1 associates with EPG-7 through a so-called FIR motif. Mutations in this motif abolish interactions in yeast two-hybrid, in a cell-free association assay, as well as in the worm. Furthermore, the authors find that ALLO-1 accumulation is partially defective in an *epg-7* mutant allele. In a last set of experiments, the authors sought to identify how IKKE-1-mediated phosphorylation orchestrates autophagy by conducting a comprehensive proteomic experiment, identifying notably a peptide of EPG-7 containing S949 as a promising lead. However, mutating this residue to alanine did not abolish paternal mitochondrial degradation.

This is a potentially interesting study that advances understanding of the mechanisms of autophagy in *C. elegans*. Although many of the conclusions are well supported by the data, some important points need to be clarified further before publication can be envisaged, as explained below.

Most important points

1. One general issue pertains to when exactly during development some of the analyses are conducted for each embryo. This is critical given that some of the effects are rather small, and that the signal intensity of GFP-ALLO-1 increases during the cell cycle. For example, in Fig. 5A-B, the authors need to make sure that the small diminution in GFP-ALLO-1 levels in the *ikke-1* null is not due simply to scoring embryos as a slightly earlier stage (related to this: please adjust brightest and contrast for the DAPI signal or else report it separately, as the current rendition does not allow one to assess developmental timing). A similar comment holds for Fig. S4A-B.

Response to 1: We thank the reviewer for the suggestion and we apologize for the unclear images. In accordance with the reviewer's comment, we have shown DAPI or DIC images to confirm the zygote stages in Fig. 5 (revised Fig. 6) and Supplementary Fig. 4 (revised Supplementary Fig. 6), and explanations of the zygotic stages have also been added to the legend. As pointed out by the reviewer, the intensity of GFP-ALLO-1a and GFP-ALLO-1b around the paternal organelles gradually increased after fertilization. A bright signal was observed during meiosis II to the pseudo-cleavage stage, and the signal seemingly started decreasing after the pronuclear meeting stage, likely because of the maturation of autophagosomes to autolysosomes (Sato et al., 2018). Therefore, we captured images during meiosis II to the pseudo-cleavage stage to compare the intensities of ALLO-1. In each figure, the wild-type and mutant zygotes at the same stage were shown for comparison.

2. The authors should quantify EPG-7-GFP distribution in the *allo-1* null allele (Fig. S3). The author write on p. 6 that the "localization... was almost completely inhibited". This is a central point because the postulated feedback mechanism schematized in Fig. 7H and featured in the title of the paper requires that NO EPG-7 or IKKE-1 be present in an *allo-1* null mutant. Otherwise parallel pathways would offer a more plausible explanation for how the system operates.

Response to 2: We thank the reviewer for your comment. In accordance with the reviewer's comment, we quantified the intensity of EPG-7-GFP and GFP-IKKE-1 around the paternal mitochondria in the *allo-1* null mutant and confirmed that no signal above the background was detected (revised Fig. 5b and Supplementary Fig. 5d). We also showed images with increased brightness (revised Supplementary Fig. 5), although we could not detect the GFP signal around the paternal mitochondria in the *allo-1* null mutant. These results suggest that EPG-7-GFP and GFP-IKKE-1 are

strongly dependent on ALLO-1.

3. The impact of IKKE-1 on ALLO-1 distribution is an important cornerstone of this study, and Fig. 3A-B clearly shows that IKKE-1 is critical for GFP-ALLO-1b recruitment to paternal mitochondria. However, no quantification is provided in Fig. S4 to make sure that the same holds for endogenous ALLO-1b. Reading the legend of the figure -which refers to a panel D that is not shown- perhaps the quantification has been conducted but somehow forgotten in the submitted version of the manuscript?

Response to 3: We thank the reviewer for the careful peer review and apologize for neglecting to include Panel D. The results of the quantification of endogenous ALLO-1b around paternal mitochondria are shown in the revised Supplementary Fig. 6f.

4. How can the authors ascertain whether GFP-LGG-1 surrounds paternal mitochondria partially or entirely without super-resolution microscopy and 3D reconstruction? Also, the authors should not only show schematics in Fig. 4D but also an exemplary image for each category so that readers can have a better sense of what the classification is based on.

Response to 4: In accordance with the reviewer's comment, we performed super-resolution microscopy and 3D reconstruction to confirm whether GFP-LGG-1 entirely enclosed the paternal organelles or partially associated them. In the wild type, GFP-LGG-1 was observed to occur spherically surrounding the paternal organelles. In contrast, in the *ikke-1* mutant, GFP-LGG-1 either partially engulfed the paternal organelle or was not localized at all. These results are consistent with those shown in revised Fig. 5c and d (original Fig. 4c and d). The representative images of this 3D reconstruction are shown in revised Fig. 5f and Supplementary Movie 3.

However, it is difficult to capture a number of 3D images for quantification using a super-resolution microscope because this technique is time-consuming. In our previous quantification, we obtained z-stacks of zygotes by confocal microscopy and assessed whether GFP-LGG-1 enclosed the paternal organelles or was only partially attached. In addition to the cartoons shown in original Figs. 4d and 5d, we have included several representative images in the revised manuscript (revised Figs. 5d and 6d). We also quantified the fluorescence intensity of GFP-LGG-1 accumulated around paternal organelles and showed that GFP-LGG-1 was less recruited in the *ikke-1*

mutant (revised Figs. 5e and 6e). This result also supports our conclusion that formation of the GFP-LGG-1-positive isolation membrane was attenuated in the *ikke-1* mutant.

Other points

5. The authors performed an impressive phosphoproteomic analysis to try to identify the relevant residue(s) phosphorylated by IKKE-1 during allophagy. Disappointingly, the S949A mutant did not seem to prevent paternal mitochondrial degradation (note, however, that there are no numbers nor quantifications in Fig. S6 to assess how firm this conclusion is). The authors should consider testing whether an S949D/E mutant can bypass the need for IKKE-1 as another way to address the potential relevance of S949 phosphorylation. Moreover, mutating all 11 phosphorylation sites could be envisaged especially since the experiments are conducted with transgenic strains.

Response to 5: In accordance with the reviewer's comment, we quantified the rescue experiment of EPG-7 S949A-GFP expressing embryos and confirmed that the S949A mutant was functional for paternal mitochondrial degradation (revised Supplementary Fig. 8c). We also created an EPG-7 mutant, in which all identified phosphorylation sites were changed to alanine; however, the mutant also significantly rescued the defect in paternal mitochondrial degradation of *epg-7(tm2508)* background (revised Supplementary Fig. 8b and c). These results suggest that paternal mitochondrial degradation is a complex mechanism involving not only EPG-7, but also the phosphorylation of other substrates by IKKE-1.

Additionally, we created an EPG-7 S949D mutant and investigated whether this mutant bypassed IKKE-1. However, this mutant did not rescue the phenotype of the *ikke-1* deletion mutant, suggesting that EPG-7 S949D was insufficient to complement the function of IKKE-1 (revised Supplementary Fig. 8d and e).

6. There appears to be a disconnect between Fig. 2B and Fig. 1. Indeed, Fig. 2B seems to indicate that most of the GFP-ALLO-1a signal (e.g. at 180'') is at paternal mitochondria, not MOs (which I assume are what the arrowheads point to, this is something the legend does not spell out), whereas this is not what Fig. 1 conveys.

Response to 6: We thank the reviewer for this suggestion and apologize for the confusing image displays. Through live imaging, ALLO-1a first appeared on paternal mitochondria and then appeared on structures outside the paternal mitochondria,

presumably MOs, approximately 3 min after sperm-oocyte contact. The fluorescence intensity of these extramitochondrial ALLO-1a puncta increased further and eventually became brighter than that of the paternal mitochondria. In revised Fig. 1c (original Fig. 1b), we showed zygotes at the meiosis II to pseudo-cleavage stage. In zygotes at this stage, ALLO-1a also localized to the paternal mitochondria, but the signal was much weaker than that of MOs, making it difficult to clearly observe the GFP-ALLO-1a signal surrounding the paternal mitochondria. In contrast, we used a relatively high laser power for live imaging to detect the initial weak ALLO-1 signal in the paternal organelles. Therefore, Figs. 1c and 3b (original Figs. 1b and 2b) may give varying impressions. To clarify these points, we have shown an image of revised Fig. 1c with the brightness adjusted to view the GFP-ALLO-1a signal surrounding the paternal mitochondria more clearly in revised Supplementary Fig. 1a. We have also shown an image of another zygote expressing GFP-ALLO-1a in Supplementary Fig. 1c and d, which also illustrates weak localization of GFP-ALLO-1a around the paternal mitochondria. We explained these results carefully in the revised manuscript. We also included explanation of arrows in the legend of revised Fig. 3.

7. The authors generated transgenic lines by bombardment for their *in vivo* experiments. A potential caveat with such lines is that expression levels can vary between strains. Insertions by MosCI would seem like a better, more controlled, strategy.

Response to 7: We recognize that this is a valid point and an issue that should be considered in the future. In the present study, we confirmed by immunoblotting that the expression levels of GFP-ALLO-1a and GFP-ALLO-b were comparable (revised Supplementary Fig. 6a).

8. To further ascertain the specificity of the ALLO-1 antibodies towards their respective isoforms, the authors could conduct Western blot analysis as the two isoforms have a different size. Moreover, they could also conduct immunofluorescence experiments with the transgenic lines expressing solely ALLO-1a or solely ALLO-1b in the background of the *allo-1* null.

Response to 8: We thank the reviewer for this comment. To compare the molecular weights of these isoforms on a single membrane, we performed immunoblotting of GFP-ALLO-1a and -b. However, we did not detect a clear difference in their sizes under our conditions (revised Supplementary Fig. 6a). The difference in size was

estimated at 1.4 kDa, which would be too small to detect using immunoblotting.

Based on this suggestion, we performed immunostaining of strains expressing GFP-ALLO-1a or GFP-ALLO-1b in an *allo-1* null mutant background to confirm the specificity of both antibodies (revised Supplementary Fig. 2c-f). GFP-ALLO-1a was not stained by the anti-ALLO-1b antibody, and vice versa. In addition, we realized that the anti-ALLO-1b antibody produced some nonspecific signals in the cytoplasm, which was observed in both the wild-type and *allo-1* null mutant backgrounds. We also performed immunoblotting of these strains and confirmed the specificity of the antibodies (revised Supplementary Fig. 3g).

9. In Fig. 2, it was not clear why the authors did not conduct such experiments also with an MO marker (bearing mCherry for instance), to ascertain that ALLO-1a truly associates with MOs.

Response to 9: We thank the reviewer for this comment. In fact, we struggled to construct a fluorescent MO marker for such experiments, but none have been successful thus far. We attempted to generate transgenic animals expressing GFP or mCherry fusions with SPE-10 and SPE-38 in sperm, but they showed no or only subtle signal of MOs in sperm and zygotes. Therefore, we confirmed the localization of endogenous ALLO-1a to MOs using immunostaining with an anti-MO antibody (1CB4). We believe that most (if not all) of the extra-mitochondrial signals in Fig. 3b (original Fig. 2b) represent ALLO-1a on the MOs.

10. Fig. S4B is lacking statistical analysis. The text on p. 7 mentions that “further accumulation was impaired”, but Fig. S4B seems to indicate little change.

Response to 10: We apologize for the confusing explanation shown in Supplementary Fig. 4b; however, Supplementary Fig. 4b (revised Supplementary Fig. 6d) shows that the GFP expression level of each strain was not different; statistical analysis of the GFP intensity around the paternal mitochondria was not shown. Therefore, a new figure showing GFP intensity accumulated around the paternal mitochondria has been added to revised Supplementary Fig. 6c.

11. The authors should indicate in the Materials and Methods what amounts of starting material (i.e. animals) were utilized for the proteomic analyses.

Response to 11: In accordance with the reviewer's comment, we have provided information on the amount of starting material for proteomic analysis in the Methods section (p. 22 and 23).

12. Table S1 does not report phosphopeptides, as the title indicates, but merely a list of proteins in which phosphopeptides were found. This needs to be fixed and the actual sequences provided.

Response to 12: We appreciate the reviewer's comments. The amino acid sequences of the detected peptides and phosphorylated residues are shown in Supplementary Table 1.

13. The authors may want to comment on whether having two isoforms of ALLO-1 is a conserved feature amongst nematodes.

Response to 13: We thank the reviewer for providing this insight. Based on WormBase, ALLO-1a is widely conserved in nematode species, whereas the ALLO-1b homolog was not detected in the current gene prediction. An ALLO-1b-like reading frame was found in the *C. briggsae* genome sequence, although its expression needs to be confirmed experimentally. Since only a limited number of ESTs have been registered in other nematode species, further analysis is required. We mentioned this point in the Discussion section as follows (p. 12):

“ALLO-1a is widely conserved in nematode species, whereas the ALLO-1b homolog was not observed in current gene prediction of WormBase. However, only a limited number of cDNA have been registered in these species, and further gene expression analyses are required.”

14. Although the precise role of IKKE-1 was unclear before this work, is this also the case of its mammalian counterparts? This should be mentioned explicitly on p. 6.

Response to 14: We thank the reviewer for this suggestion. We have mentioned the recent findings on TBK1, the mammalian homolog of IKKE-1, as follows on p. 4;

“In mammals, TRAF-associated NF- κ B activator-binding kinase 1 (TBK1) is required for several selective autophagy pathways, including the PTEN-induced kinase 1

(PINK1)–Parkin-mediated mitophagy. TBK1 phosphorylates several autophagy-related factors such as OPTN, NDP52, and p62, and promotes binding to LC3 or ubiquitin.^{24, 25, 26, 27} Additionally, TBK1-mediated phosphorylation of LC3 family proteins or Rab7A is involved in PINK1–Parkin dependent mitophagy.^{28, 29} A recent study also suggested that TBK1 directly binds to class III phosphatidylinositol 3-kinase to initiate autophagy in the OPTN-mediated mitophagy, whereas it functions redundantly with ULK1/2 in the NDP52-mediated pathway.³⁰ Thus, TBK1 has been suggested to play multiple roles; however, the mechanism by which TBK1 regulates selective autophagy is not fully understood, particularly in animal models and systems other than mammals.”

Small points

15. On p. 5, the authors mention that they “developed” a method for time-lapse imaging of allophagy, but reading the corresponding Materials and Methods section did not make it clear what aspects -beyond standard live imaging approaches- were developed specifically for this purpose.

Response to 15: We appreciate the reviewer’s comment and have added an explanation of how our live imaging method varies from existing imaging methods and its advantages in the Methods section (p. 22).

16. Fig. S2: note that pronuclei do not fuse in *C. elegans* when they meet, they merely appose. Also, therefore, the schematic in mitosis should represent two pronuclei, not just one, and the text also be rectified in a couple of places accordingly (on p. 6 for instance). Also, at what temperature is the time line? This should be indicated.

Response to 16: We thank the reviewer for this helpful comment. We have changed all “pronuclear fusion” to “pronuclear meeting”. In addition, we have added temperature information to the legend of revised Fig. 1a and Supplementary Fig. 4.

17. On p. 7, the authors write that “... GFP-LGG-1 puncta were occasionally attached to 23-32% of paternal mitochondria...”. This is confusing: grammatically, this seems to indicate that occasional attachment was observed in these 23-32%, but what about the others? Or do the authors mean to convey that attachment was observed in this percentage of embryos? Please clarify.

Response to 17: We thank the reviewer for this comment. To clarify the meaning of the percentage, we have added a detailed explanation on p. 8 as follows:

“In the *ikke-1* deletion mutants (two different alleles were used), the GFP-LGG-1 signal was undetectable in approximately 68–77% of paternal mitochondria observed, suggesting that the partial accumulation of ALLO-1 and EPG-7 in the *ikke-1* mutant was insufficient to initiate LGG-1-positive isolation membrane formation. However, localization of small GFP-LGG-1 puncta was occasionally observed in the remaining 23–32% of paternal mitochondria in these *ikke-1* mutants. This partial localization of GFP-LGG-1 occurred in the kinase-dead mutant of *ikke-1* (Fig. 5c and d).”

18. When the authors write on p. 10 that “ALLO-1 is the most upstream regulator,” they should write instead “ALLO-1 is the most upstream regulator known,” as other component yet to be discovered might take on that role in the future.

Response to 18: We agree with the reviewer’s suggestion and have added “among known factors” on p. 10.

19. What is an FV1200 microscope system (p. 18)? The authors should provide more explanation.

Response to 19: Thank you for your comment. We have added an explanation in the Methods section (p. 20).

20. On p. 19, the authors list three lenses with which time-laps imaging was conducted apparently. If so, the term “or” seems to be missing between the mention of the lenses.

Response to 20: We thank the reviewer for this careful peer review. We have added “or” in the relevant part of the article (p. 22).

21. Throughout the figures: by convention, embryos should be shown with the anterior to the left.

Response to 21: We appreciate the reviewer's comments. We have corrected the orientation of the embryos shown in all Figures.

22. Fig. 7F–G: the legend is missing on the X axis.

Response to 22: We thank the reviewer for this comment. In accordance with this comment, we have added an explanation of the X-axis of revised Fig. 8f and g in the legend.

23. Fig. S5 Type: H. sapiens (no capital).

Response to 23: This error in Supplementary Fig. 5a (revised Supplementary Fig. 7a) has been corrected in accordance with the reviewer's comment.

REVIEWERS' COMMENTS

Reviewer #1 (Remarks to the Author):

The authors have addressed all my concerns. I thus recommend publication of the paper.

Reviewer #2 (Remarks to the Author):

The authors' responses are appropriate and I have no further comments. Please just confirm the following points

- Fig. 1K does not appear to be referred to in the text.
- The rebuttal letter states that phosphorylation of ALLO-1 is shown in Supplementary Table 1, but such an addition seems not found.

Reviewer #3 (Remarks to the Author):

The authors have sufficiently answered all questions and concerns. No further revisions are requested. I am happy that the authors have made sufficient effort to address my comments.

Reviewer #4 (Remarks to the Author):

The authors have addressed the vast majority of the concerns of the four reviewers in a satisfactory manner, resulting in a substantially improved manuscript. Publication is recommended after the authors address the following remaining minor points.

- Prompted by my suggestion (#5 in the rebuttal document), the authors generated a variant of EPG-7 with all 13 IKKE-1-mediated S/T phosphorylation sites mutated to A, and yet found that had no impact on allophagy. Moreover, their newly generated S949D EPG-7 mutant did not bypass the requirement for IKKE-1. Although it remains possible that additional unidentified phosphorylation sites on EPG-7 are important, as the authors mention, perhaps a more parsimonious explanation is that the relevant substrate for IKKE-1 phosphorylation is not EPG-7. This possibility must be spelled out explicitly on p. 10

of the revised manuscript and further addressed in the discussion.

- I remain doubtful that the addition of silicon sheets for time-lapse imaging (as now reported on p. 23 of the revised manuscript) qualifies for a “development” as heralded on p. 5, but I leave it to the authors to decide whether to keep the wording as such or not.

- In response to one of my comments, the authors rectified the wording regarding pronuclear meeting, and changed some of the schematics. However, two further minor changes must be made to Fig. 1a and Fig. S4: 1) the pronuclei meet in the posterior of the embryo, not in the anterior as currently represented in the Pseudocleavage and Pronuclear meeting stages; 2) as mentioned already in my initial review, two pronuclei, not just one, should be shown at Mitosis. Also, in the first inset of Fig. 1a, “Paternal nuclei” should read “Paternal nucleus”.

- Legend of Fig. S1a: “Fig. 1b” should read “Fig. 1c”. Also, please indicate explicitly that these are allo-1(t4756) embryos (as done in Fig. 1).

Response to reviewers' comments

REVIEWERS' COMMENTS

Reviewer #2 (Remarks to the Author):

The authors' responses are appropriate and I have no further comments. Please just confirm the following points

- Fig. 1K does not appear to be referred to in the text.

We thank the reviewer for the careful review. We have cited Fig. 1k in the first paragraph of the Results section (p5).

- The rebuttal letter states that phosphorylation of ALLO-1 is shown in Supplementary Table 1, but such an addition seems not found.

We apologize for our careless mistake. We have included this information as new Supplementary Table 2.

Reviewer #4 (Remarks to the Author):

The authors have addressed the vast majority of the concerns of the four reviewers in a satisfactory manner, resulting in a substantially improved manuscript. Publication is recommended after the authors address the following remaining minor points.

- Prompted by my suggestion (#5 in the rebuttal document), the authors generated a variant of EPG-7 with all 13 IKKE-1-mediated S/T phosphorylation sites mutated to A, and yet found that had no impact on allophagy. Moreover, their newly generated S949D EPG-7 mutant did not bypass the requirement for IKKE-1. Although it remains possible that additional unidentified phosphorylation sites on EPG-7 are important, as the authors mention, perhaps a more parsimonious explanation is that the relevant substrate for IKKE-1 phosphorylation is not EPG-7. This possibility must be spelled out explicitly on p. 10 of the revised manuscript and further addressed in the discussion.

We agree with the referee's comment. According to the referee's comment we have mentioned this possibility in the Results and Discussion sections as follows:

“Therefore, phosphorylation of multiple autophagy regulators other than EPG-7 by IKKE-1 may be involved in ALLO-1 accumulation.” (p10 in the Results section)

“However, no obvious allophagy defects were observed in the phosphorylation site mutants. Therefore, it is possible that IKKE-1-dependent phosphorylation sites of EPG-7, which were not detected in this analysis, or IKKE-1-dependent phosphorylation of autophagy factors other than EPG-7 may enhance or stabilize the interaction between EPG-7 and ALLO-1 and contribute to ALLO-1 accumulation.” (p12 in the Discussion section)

- I remain doubtful that the addition of silicon sheets for time-lapse imaging (as now reported on p. 23 of the revised manuscript) qualifies for a “development” as heralded on p. 5, but I leave it to the authors to decide whether to keep the wording as such or not.

We have withdrawn this statement. We mentioned our methods in the Methods section as follows:

“We improved a method to immobilize adult hermaphrodites for live imaging.”

- In response to one of my comments, the authors rectified the wording regarding pronuclear meeting, and changed some of the schematics. However, two further minor changes must be made to Fig. 1a and Fig. S4: 1) the pronuclei meet in the posterior of the embryo, not in the anterior as currently represented in the Pseudocleavage and Pronuclear meeting stages; 2) as mentioned already in my initial review, two pronuclei, not just one, should be shown at Mitosis.

We thank the reviewer for this helpful comment. We have revised Fig. 1a and Fig. S4.

- 1) The pronuclei migrate to and meet in the posterior of embryos.
- 2) We also showed two pronuclei at the Mitosis stage.

Also, in the first inset of Fig. 1a, “Paternal nuclei” should read “Paternal nucleus”.

We have corrected it to “Paternal nucleus”.

- Legend of Fig. S1a: “Fig. 1b” should read “Fig. 1c”. Also, please indicate explicitly that these are allo-1(t4756) embryos (as done in Fig. 1).

We have changed “Fig. 1b” to “Fig. 1c”. We also indicated that GFP-ALLO-1a was expressed in the *allo-1(tm4756)* background in the figure as well as in the figure legend.